# MA²E: Addressing Partial Observability in Multi-Agent Reinforcement Learning with Masked AutoEncoder

**Sehyeok Kang**[1*] **Yongsik Lee**[1*] **Gahee Kim**[1] **Song Chong**[1†] **Se-Young Yun**[1†]
KAIST AI[1]
{kangsehyeok0329,dldydtlr93,gaheekim,songchong,yunseyoung}
@kaist.ac.kr

## Abstract

Centralized Training and Decentralized Execution (CTDE) is a widely adopted paradigm to solve cooperative multi-agent reinforcement learning (MARL) problems. Despite the successes achieved with CTDE, partial observability still limits cooperation among agents. While previous studies have attempted to overcome this challenge through communication, direct information exchanges could be restricted and introduce additional constraints. Alternatively, if an agent can infer the global information solely from local observations, it can obtain a global view without the need for communication. To this end, we propose the Multi-Agent Masked Auto-Encoder (MA²E), which utilizes the masked auto-encoder architecture to infer the information of other agents from partial observations. By employing masking to learn to reconstruct global information, MA²E serves as an inference module for individual agents within the CTDE framework. MA²E can be easily integrated into existing MARL algorithms and has been experimentally proven to be effective across a wide range of environments and algorithms. The code is available at https://github.com/cheesebro329/MA2E.

## 1 Introduction

In cooperative multi-agent tasks, the environments are typically partially observable, where individual agents do not have complete access to the global information. For instance, consider a motivating scenario where allied and enemy forces are engaged in combat as illustrated in Figure 1. An ally agent 1 might perceive the current situation as favorable by observing only one enemy, while ally agents 2 and 3 observe four and five enemies, respectively, and therefore assess the situation as unfavorable. This discrepancy makes it difficult for the agents to make cooperative decisions. Therefore, addressing partial observability is a crucial challenge in the field of cooperative Multi-Agent Reinforcement Learning (MARL) (Nguyen et al., 2020; Canese et al., 2021; Ning & Xie, 2024).

One straightforward approach is fully centralized training and control of a joint policy for all agents which simplifies the MARL problem into a single-agent one. However, the centralized setting suffers from scalability and heavy computational costs as the number of agents increases due to the curse of dimensionality (Gronauer & Diepold, 2022; Du & Ding, 2021; Canese et al., 2021).

Centralized Training and Decentralized Execution (CTDE) paradigm provides a structured framework for mitigating aforementioned issues (Canese et al., 2021; Ning & Xie, 2024; Du & Ding, 2021; Gronauer & Diepold, 2022), but still fails to fully resolve the problem of partial observability. Although the global information is used during training, each agent relies solely on its local observations during execution. Such discrepancies between learning and execution can limit collaborative decision-making (Shao et al., 2022; Yuan et al., 2022; Guan et al., 2022).

Previous works have explored the strategies to relieve drawback of decentralized execution. Relaxing CTDE with communication by allowing direct messaging among agents have been extensively studied, but it faces limitations in real-world scenarios where inter-agent communication is not permitted or is

---

*Equal contribution. †Corresponding authors.

constrained by various factors such as limited bandwidth and noisy channels (Zhu et al., 2022; Ning & Xie, 2024). Without communication, shared structures such as common knowledge (Schroeder de Witt et al., 2019) or an one-hot consensus (Xu et al., 2023) have been proposed to foster cooperation. However, such abstractly encoded representations may be lossy in complex environments.

Masked modeling, which learns to reconstruct original data from masked input, has shown notable achievements in the language (Devlin et al., 2018; Liu, 2019; Clark, 2020), the vision (Chen et al., 2020; Bao et al., 2021; He et al., 2022), and even in the single-agent reinforcement learning (RL) domains (Seo et al., 2023; Liu et al., 2023a; Wu et al., 2023). Inspired by those successes, in this paper, we propose Multi-Agent Masked AutoEncoder (MA²E) which enables agents to infer the global information based solely on partial observations. Our key idea is to use a masked autoencoder (MAE) (He et al., 2022), which masks a subset of the input and is trained to recover the missing (masked) data using the visible (unmasked) regions, to reconstruct full trajectories of all agents from local observations. While such masked modeling usually targets to solve the downstream tasks with complete data after training from incomplete input, our focus is to

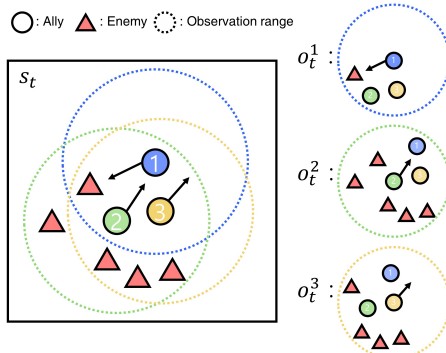

Figure 1: A motivating combat scenario. The solid circles and red triangles denote the ally agents and enemies, respectively. The dotted circles are observation ranges of each ally agent. While the current global state $s_t$ is identical, each agent perceives the situation differently due to the discrepancy among their partial observations $o_t^1, o_t^2, o_t^3$.

utilize MAE's capability to handle partially observable data to infer the entire information. In MA²E, MAE performs masking on a per-agent basis and reconstructs the original global input, allowing the inference of global information from partial observations. This enhances the agent's ability to deduce the situations of other agents and aids in making appropriate decisions. Furthermore, by separately training MA²E from agents' policies, MA²E can be easily plugged into existing MARL algorithms as backbone networks and the learned MAE can be transferred to other backbone policies.

We experimentally evaluate our approach on the Starcraft Multi-agent Challenge (SMAC) (Samvelyan et al., 2019), SMACv2 (Ellis et al., 2023), and Google Research Football (GRF) (Kurach et al., 2020) environments. The experimental results consistently demonstrate that MA²E achieves faster convergence and higher sample efficiency compared to fine-tuned QMIX (Hu et al., 2021), which is the state-of-the-art MARL algorithm. Additionally, MA²E shows comparable or superior performance compared to the cases where full observations are provided or communication is employed, substantiating the ability of MA²E to effectively infer full observations from partial observations. Furthermore, employing MA²E achieves performance gains for various value-based and policy-based MARL methods as a backbone algorithm. Finally, to effectively integrate MA²E into the backbone MARL algorithms, we propose an appropriate configuration for applying MA²E across different hyperparameters and masking strategies in ablation studies.

## 2 RELATED WORK

**Partial Observability in MARL**: Partial observability is a fundamental challenge in cooperative MARL. As a naïve adoption of a fully centralized setting results in an intractable computational complexity (Canese et al., 2021; Du & Ding, 2021; Gronauer & Diepold, 2022), alternative strategies have been explored. Centralized Training and Decentralized Execution (CTDE) is a popular framework to solve partial observability (Canese et al., 2021; Ning & Xie, 2024; Du & Ding, 2021; Gronauer & Diepold, 2022). In value-based CTDE algorithms, value decomposition which factorizes a joint value function into individual ones is dominant (Sunehag et al., 2017; Rashid et al., 2018; Lowe et al., 2017; Foerster et al., 2018; Yu et al., 2022). Another major branch is policy-based CTDE methods, where actor-critic approach is widely-adopted (Lowe et al., 2017; Foerster et al., 2018; Yu et al., 2022). We aim to alleviate the limitation of fully decentralized execution solely with local observations while following the CTDE paradigm. Subsequent to some pioneer works (Sukhbaatar et al., 2016; Foerster et al., 2016), communication among agents has been extensively studied in cooperative MARL (Yuan

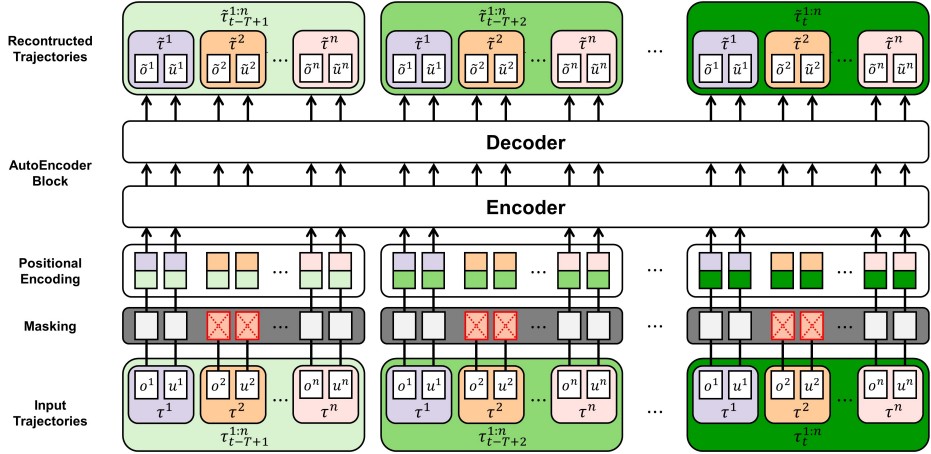

Figure 2: The architecture of MA$^2$E. During centralized training, MA$^2$E masks out $k$ agents' trajectories from all agents' trajectories $\tau^n_{t-T+1:t}$ and learns to reconstruct them. The positional encoding is applied considering both the time and the agent information. During decentralized execution, trajectories of other agents are masked except the trajectories of current agent.

et al., 2022; Hu et al., 2021; 2024). Although communication can mitigate partial observability, it may not be possible in target environments or constrained by diverse factors (Zhu et al., 2022; Ning & Xie, 2024). Li et al. (2023) gradually shifts from communication to tacit collaboration, but it requires communication during exploration. Our goal is to remove the dependence on communication. Another line of research is to utilize shared structures dependent only on local information during execution. MACKRL (Schroeder de Witt et al., 2019) employs common knowledge among groups of agents with a hierarchical policy tree. COLA (Xu et al., 2023) infers the same one-hot consensus for all agents from different local observations. Compared to those limited abstractions, our method enables each agent to recover the global trajectories of all agents from its local observations.

**Masked Modeling in Reinforcement Learning**: Masked modeling to reconstruct original data from masked ones emerges as a powerful technique in diverse domains including vision (Chen et al., 2020; Bao et al., 2021; He et al., 2022) and NLP (Devlin et al., 2018; Liu, 2019; Clark, 2020). Recently, the idea has been extended to the RL field. Masked World Model (Seo et al., 2023) is trained to reconstruct pixels from masked convolutional features to learn a latent dynamics model. MaskDP (Liu et al., 2023a) and MTM (Wu et al., 2023) apply masking to a portion of input trajectories and learn to reconstruct them. They show generalization abilities on diverse tasks by manipulating masking patterns desired for the target task. Aforementioned methods aim to solve the single-agent RL tasks. In contrast, we explore MARL problems with masked modeling. MA2CL (Song et al., 2023) deploys masked modeling with contrastive loss. The policy and the reconstruction module are jointly trained to obtain encoder representation for the policy with enhanced collaboration. In contrast, MA$^2$E separately trains the policy and the reconstruction module, and its main focus is to recover global information rather than encoder representation. MaskMA (Liu et al., 2023b) treats MARL as a sequence modeling problem and masking is used to predict next action, rather than reconstructing the inputs. In contrast, our aim is to recover global information using MAE for decentralized execution.

## 3 METHOD

**Problem Formulation**: We consider a cooperative MARL task which can be described as a Dec-POMDP (Oliehoek et al., 2016). Dec-POMDP is defined as a tuple $G = \langle S, U, P, r, Z, O, n, \gamma \rangle$ where $s \in S$ denotes the global environment state. At each time step, each agent $i \in \{1, ..., n\}$

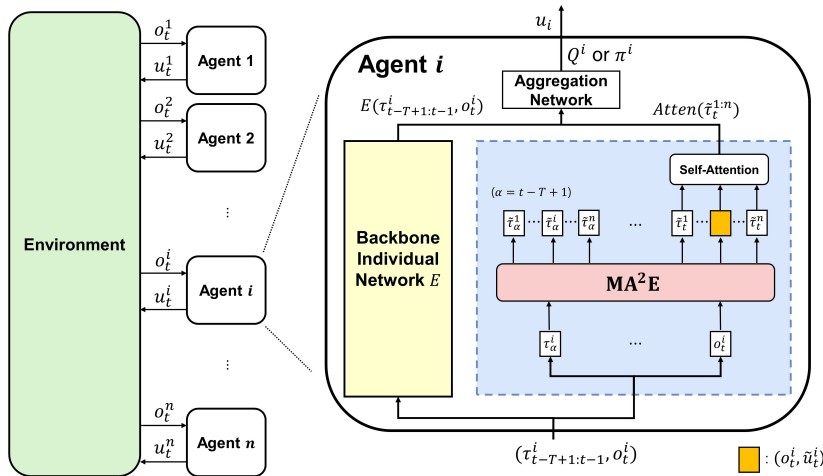

Figure 3: Incorporating MA$^2$E into individual agents. Given a backbone MARL algorithm, MA$^2$E is integrated into the backbone individual network of each agent during decentralized execution. MA$^2$E serves as a global information inference module by only using local observations of the current agent and inferring other agents' information.

chooses an action $u^i \in U$ which consists a joint action $\mathbf{u} = \{u^1, \cdots, u^n\} \in U^n$. The environment follows the transition function $P(s'|s, \mathbf{u}) : S \times U^n \times S \to [0, 1]$. All the agents share the common reward function $r(s, \mathbf{u}) : S \times U^n \to \mathbb{R}$. Each agent only obtains an individual partial observation $o^i \in Z$ with the observation function $O(s, i) : S \times N \to Z$, and has an observation-action history $\tau^i \in T \equiv (Z \times U)^*$ and an individual policy $\pi^i(u^i|\tau^i) : T \times U \to [0, 1]$. The objective is to maximize the expected return $\mathbb{E}_{s_{t+1:\infty}, \mathbf{u}_{t+1:\infty}}[\sum_{m=0}^{\infty} \gamma^m r_{t+m}|s_t, \mathbf{u}_t]$ with a discount factor $\gamma \in [0, 1)$.

**Masking**: While random masking to states and actions is commonly adopted in single agent RL (Liu et al., 2023a; Wu et al., 2023), our aim is to infer other agents' information only with local observations hence we apply masking at the agent level. During centralized training, we can access all agents' trajectories $\tau^n_{t-T+1:t} = \{\tau^n_{t-T+1}, \ldots, \tau^n_t\}$ where $\tau^i_t = (o^i_t, u^i_t)$. As illustrated in Figure 2, we randomly select $k$ agents and mask out all trajectories of these agents. Then MA$^2$E learns to recover the entire trajectories from the masked trajectories that it outputs $\tilde{\tau}^{1:n}_{t-T+1:t}$ where $\widetilde{\tau}^i_t$ represents a recovered trajectory for the agent $i$ at time step $t$.

In decentralized execution (for exploration during training or deployment after training), apart from the local observations and actions of the agent $(\tau^i_{t-T+1:t-1}, o^i_t)$, other information such as observations from other agents cannot be obtained. The agent only utilizes its local observations for the action selection and MA$^2$E infers global information from local observations by restoring masked areas during execution, as depicted in Figure 3. The comparison between agent level and random masking strategies can be found in Section 4.4 and Appendix D.

**Architecture**: The detailed architecture of MA$^2$E is described in Figure 2. MA$^2$E follows an auto-encoder architecture (Bank et al., 2020), with the encoder and decoder structured similarly to Transformer (Vaswani et al., 2017), using Multi-Head Attention (MHA) and feedforward networks. The decoder does not use masked MHA. Instead, a separate layer is utilized for masking before the input data is fed into the encoder and decoder. In other words, observations and actions are embedded in the embedding layer, undergo masking and positional encoding layers, then are fed into the encoder and decoder. For positional encoding, the orders of both time steps and agents are valuable information as MA$^2$E takes histories of multiple agents. Therefore, we apply positional encoding considering *Agent* and *Time* dimensions, and the related results are presented in Section 4.4 and Appendix D.

**Incorporating MA$^2$E into Individual Agent**: MA$^2$E is integrated into individual agents as illustrated in Figure 3. Following this procedure, we note that MA$^2$E can be easily plugged into existing MARL algorithms. Depending on the selected MARL algorithm, each agent has a backbone network $E$ corresponding to the value function $\bar{Q}^i$ or policy $\bar{\pi}^i$. MA$^2$E is integrated internally within each agent

in addition to $E$. Each agent only takes its own local observations to determine actions; hence, inputs for both $E$ and MA$^2$E are also local observations. Then MA$^2$E reconstructs trajectories of all agents from its local trajectories, where the output for the agent $i$ at the current time $t$ is replaced by $(o_t^i, \bar{u}_t^i)$ since $o_t^i$ is already observed. The recovered trajectory for the current time $t$, $\widetilde{\tau}_t^{1:n} = (\widetilde{\tau}_t^1, ..., \widetilde{\tau}_t^n)$, is fed into the self-attention layers. The self-attention layers use a self-attention mechanism (Vaswani et al., 2017) to selectively focus on other agents' information for the agent's current situation $\widetilde{\tau}_t^i$. For the agent $i$ and $j$, the attention weight $w_{i,j}$ is calculated as:

$$w_{i,j} = \frac{\exp(\widetilde{\tau}_i^T W_k^T W_q \widetilde{\tau}_j)}{\sum_{j=1,\cdots,i-1}^{i+1,\cdots,n} \exp(\widetilde{\tau}_i^T W_k^T W_q \widetilde{\tau}_j)} \tag{1}$$

$W_q$ and $W_k$ are weight matrices such as fully connected layers for the query $q$ and and key $k$, respectively. Then a relevance-weighted value is computed by taking the weighted sum of the attention weights and values as shown below:

$$Atten_i(\widetilde{\tau}_t^{1:n}) = \sum_{j=1,\cdots,i-1}^{i+1,\cdots,n} w_{i,j} W_v \widetilde{\tau}_t^j, \tag{2}$$

where $W_v$ is another weight matrix. From the backbone network $E$, we take the outputs before the last layer $E(\tau_{t-T+1:t-1}^i, o_t^i)$ as latent variables, and both $E(\tau_{t-T+1:t-1}^i, o_t^i)$ and $Atten_i(\widetilde{\tau}_t^{1:n})$ are fed into the aggregation network. Finally, the aggregation network produces the individual value $Q^i$ or policy $\pi^i$ from which the agent chooses the action $u^i$. This framework enables fully decentralized execution for each agent without access to global information.

**Training**: The training of MA$^2$E is divided into two stages: MA$^2$E pre-training and MA$^2$E fine-tuning. During the pre-training stage, MA$^2$E is trained using samples collected by a random policy before the policy training begins. If we train MA$^2$E from scratch and agents' policy concurrently, the generated information from MA$^2$E would be a noise that interferes with the policy training. Hence, the pre-training is needed to ensure that MA$^2$E can properly infer unobserved information. The objective of MA$^2$E, defined in Eq. (3), is to minimize the Mean Squared Error (MSE) loss between masked and true histories. Pre-training continues until the loss falls below a specified threshold.

$$L_{\text{MA}^2\text{E}} = \frac{1}{nT} \sum_{t=1}^{T} \sum_{i=1}^{n} (\tau_t^i - \widetilde{\tau}_t^i)^2. \tag{3}$$

After pre-training of MA$^2$E, the policy training begins. In this stage, fine-tuning of MA$^2$E also takes place. However, the policy is independently trained according to the chosen MARL method, and MA$^2$E is periodically updated with the loss defined in Eq. (3) after updating the policy sufficient times. Both the agents' policy and MA$^2$E are trained using collected trajectories in a replay buffer.

## 4    EXPERIMENT AND RESULT

**Environment and Setup** We evaluate MA$^2$E on the StarCraft Multi-agent Challenge (SMAC) (Samvelyan et al., 2019), SMACv2 (Ellis et al., 2023), and Google Research Football (GRF) (Kurach et al., 2020) environments. SMAC is one of the most popular MARL benchmark, which covers a wide range of cooperative microcontrol scenarios. We conduct the experiments on SMAC `HARD` and `SuperHARD` scenarios. SMACv2 complements the deterministic property of SMAC by randomizing start positions and unit types, and changing the units' sight and attack ranges. In GRF, multiple agents cooperate to play a football game. All the experiments are conducted during $2 \times 10^6$ time steps for each run, and we report the average win rates with the shaded standard error from three different random seeds. More experimental details can be found in Appendix A.

### 4.1    IMPROVING PERFORMANCE BY INCORPORATING MA$^2$E INTO MARL ALGORITHMS

Firstly, we investigate the effectiveness of integrating MA$^2$E into existing MARL methods to improve the performance. We employ a fine-tuned QMIX (Hu et al., 2021), which achieves SOTA performance through parameter fine-tuning from the vanilla QMIX (Rashid et al., 2018), as a backbone algorithm to apply MA$^2$E. We compare MA$^2$E with various MARL algorithms including VDN (Sunehag et al.,

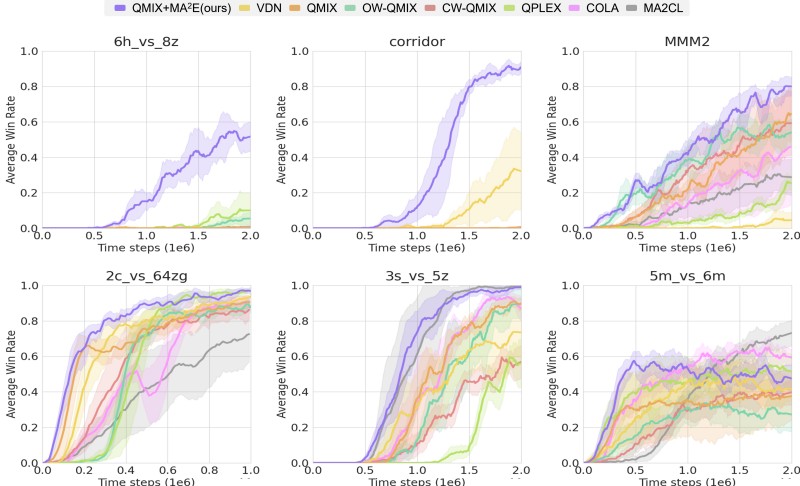

Figure 4: Performance comparison with baselines in SMAC scenarios. The blue line represents the model where MA$^2$E is plugged into QMIX, while the other colored lines correspond to baselines. The proposed model performs better in both `HARD` and `SuperHARD` scenarios in SMAC.

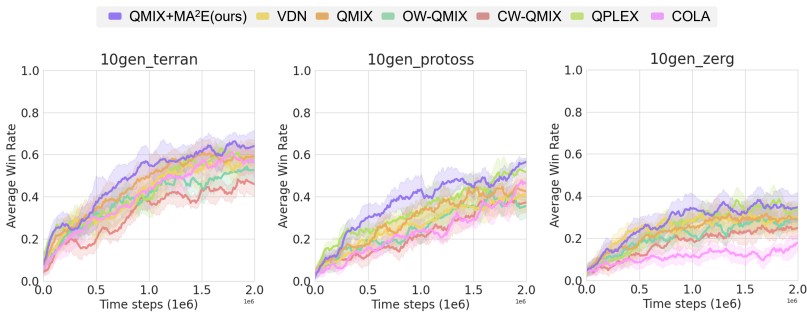

Figure 5: Performance comparison with baselines in SMACv2 scenarios. The blue line represents the model where MA$^2$E is plugged into QMIX, while the other colored lines correspond to baselines.

2017), QMIX (Rashid et al., 2018), OW-QMIX (Rashid et al., 2020), CW-QMIX (Rashid et al., 2020), QPLEX (Wang et al., 2020), MAC2L(Song et al., 2023), COLA(Xu et al., 2023) as baselines.

The performance comparison between the model with MA$^2$E and the baselines is illustrated in Figure 4 and Figure 5. **QMIX+MA$^2$E** is our proposed model, which is a fine-tuned QMIX (Hu et al., 2021) with the addition of MA$^2$E. In the `HARD` and `SuperHARD` scenarios of SMAC and in SMACv2, QMIX+MA$^2$E exhibits higher sample efficiency and superior win rates compare to all baselines. In scenarios with higher difficulty levels, such as `corridor` and `6h_vs_8z`, the performance difference between the proposed model and the baselines becomes more pronounced compared to relatively easier scenarios like `2c_vs_64zg`. Specifically, in challenging scenarios such as `corridor` and `6h_vs_8z`, baselines struggle to achieve victories even after two million time steps, whereas QMIX+MA$^2$E attains high win rates. In addition, even when compared to QMIX, which is a fine-tuned version showing state-of-the-art performance in MARL, the proposed method shows superior performance. Even in the `10gen_terran` scenario in SMACv2, QMIX+MA$^2$E demonstrates higher performance compared to the baselines.

## 4.2 COMPARISON WITH THE FULL STATE AND COMMUNICATION METHODS

We evaluate whether MA$^2$E can accurately infer full observations from partial observations. We compare the results of the model when using full state with the results of the model using MA$^2$E.

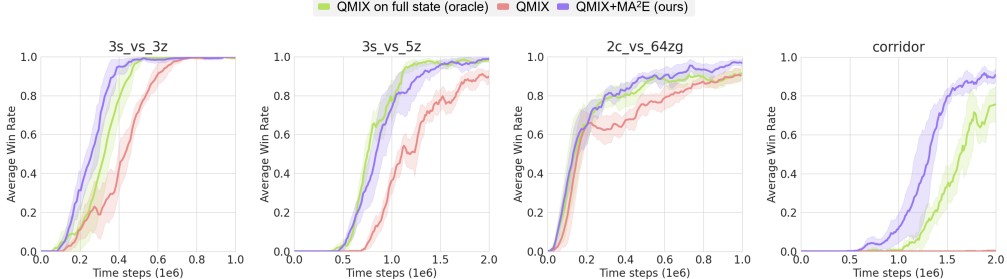

Figure 6: Performance comparison between using full observation and utilizing MA$^2$E in different SMAC scenarios. Green line represents the case where QMIX is used with the full state, the blue line corresponds to our proposed model, and the red line represents the fine-tuned QMIX which is a baseline model in the experiment. The win rates of the model using the full state and the model with MA$^2$E applied exhibit similar patterns.

Table 1: Comparison of MA$^2$E observation inference accuracy based on training progress

| Type | Before Pre-Training MA$^2$E | After Pre-Training MA$^2$E | After Fine-Tuning MA$^2$E |
|---|---|---|---|
| ● : True Allies 
 ▲ : True Enemies 
 ● : Inferred Allies 
 ▲ : Inferred Enemies | | | |
| **Discrepancy** | | | |
| Distance (L2 norm) | 0.9933 | 0.3989 | 0.3588 |
| Health (L1 norm) | 0.6606 | 0.5585 | 0.4766 |
| All Info (L1 norm) | 0.5661 | 0.3984 | 0.3584 |

The fully observable model replaces the MA$^2$E of individual agents which is Figure 3 with full observation and utilizes the output obtained through the self-attention layer. The results are depicted in Figure 6. The comparison across four SMAC scenarios shows a similar pattern between using full observation and using MA$^2$E. Moreover, it is evident that the performance is better when using full observation or using MA$^2$E compared to using only partial observations. Especially in scenarios like 3s_vs_5z, where inter-agent information sharing is crucial and the difficulty is higher compared to other scenarios, the differences become more pronounced.

Table 1 compares the inference accuracy according to the training progress of MA$^2$E. It shows the results of agent 1 inferring the observations of agent 2 in the 3s_vs_5z scenario in SMAC. From left to right in the table, the results are from models at progressively advanced stages of training, and the figure displays the inferred relative positions of agents from the observation inference results of other agents, plotted on a two-dimensional plane. In the figure, red and blue icons represent true values and the inferred values, respectively, indicating that as learning progresses, the inferred values become closer to the actual values. In addition, the values in the table mean the differences between the actual values and the MA$^2$E's deduced values for position, health, and the entire observation values of other agents. The discrepancies decrease as MA$^2$E learns, demonstrating that MA$^2$E can infer values closer to the real values through training. Based on the results, we can conclude that MA$^2$E can successfully infer information similar to full states.

Another way to utilize information over partial observations is information exchange among agents through communication. We compare MA$^2$E with communication methods in MARL domain including MAIC (Yuan et al., 2022), QMIX-att (Hu et al., 2021) and CommFormer (Hu et al., 2024). The learning curves can be found in Appendix E. As shown in Table 2, QMIX+MA$^2$E demonstrates performance that is comparable to or better than communication methods. Since MA$^2$E does not rely on direct information exchange, its performance highlights the strengths of our method.

Table 2: Comparison of MA$^2$E with communication-based methods

| Scenario | Steps | QMIX+MA$^2$E (ours) | MAIC | QMIX-att | CommFormer |
|---|---|---|---|---|---|
| 3s_vs_5z | 2M | **0.99 ± 0.01** | 0.86 ± 0.05 | 0.41 ± 0.37 | 0.06 ± 0.04 |
| 2c_vs_64zg | 1M | **0.97 ± 0.03** | 0.76 ± 0.18 | 0.34 ± 0.31 | 0.59 ± 0.06 |
| 5m_vs_6m | 2M | 0.48 ± 0.08 | 0.69 ± 0.07 | **0.81 ± 0.04** | 0.61 ± 0.05 |
| corridor | 2M | **0.91 ± 0.10** | 0.0 ± 0.0 | 0.30 ± 0.23 | 0.0 ± 0.0 |
| 6h_vs_8z | 2M | **0.53 ± 0.08** | 0.0 ± 0.0 | 0.14 ± 0.04 | 0.0 ± 0.0 |
| MMM2 | 2M | 0.79 ± 0.08 | **0.97 ± 0.07** | 0.20 ± 0.08 | 0.0 ± 0.0 |

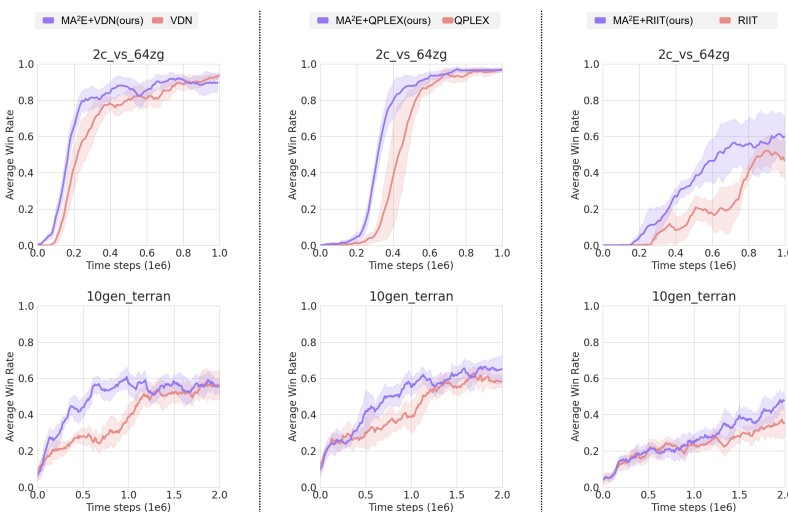

Figure 7: Performance comparison between the model incorporating MA$^2$E in QPLEX and VDN and baselines in SMAC and SMACv2 scenarios. **(Left)**: Comparing the performance of the model with MA$^2$E applied to VDN and the performance of VDN. **(Middle)**: Comparing the performance of the model with MA$^2$E applied to QPLEX and the performance of QPLEX. **(Right)**: Comparing the performance of the model with MA$^2$E applied to RIIT and the performance of RIIT to assess whether MA$^2$E leads to performance improvement when applied to a policy-based method.

## 4.3 EXTENSIBILITY AND TRANSFERABILITY OF MA$^2$E

To explore the applicability of MA$^2$E to various MARL algorithms in a general context, we conduct experiments by applying MA$^2$E to various value-based models such as VDN and QPLEX. The experimental results are as shown in the Figure 7. The results across scenarios in both SMAC and SMACv2 indicate that the addition of MA$^2$E improves performance compared to the baselines. MA$^2$E not only enhances sample efficiency but also converges to higher win rates in fewer time steps.

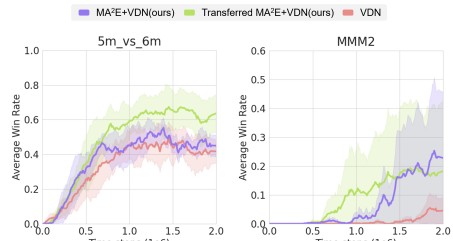

Figure 8: Performance comparison between transferred MA$^2$E and default MA$^2$E in different SMAC scenarios.

Furthermore, to confirm the applicability of MA$^2$E to policy-based algorithms, we conduct experiments by incorporating MA$^2$E into the policy-based algorithm RIIT (Hu et al., 2021). The right figures in Figure 7 illustrate that RIIT with MA$^2$E exhibits better performance compared to the baseline models. The experimental results demonstrate that MA$^2$E can be seamlessly integrated into both value-based and policy-based algorithms.

In order to confirm whether MA$^2$E can be transferred to other backbone networks, MA$^2$E is first trained with QMIX. Then the trained MA$^2$E is transferred to the VDN network and only MA$^2$E

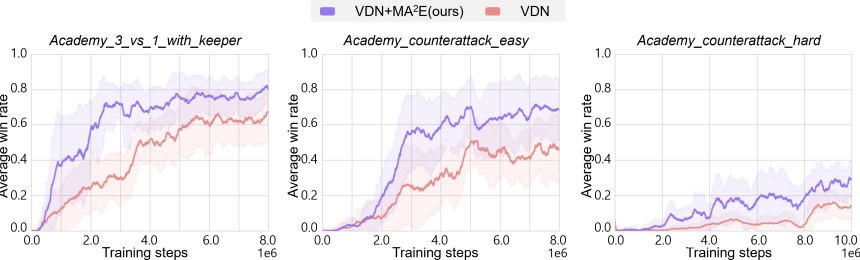

Figure 9: Performance comparison between the model incorporating MA$^2$E in VDN and baselines in Google Research Football scenarios

finetuning is performed as training progresses, without pre-training. Figure 8 compare the transferred MA$^2$E with the MA$^2$E that goes through the pre-training process. The model equipped with transferred MA$^2$E performs better in the `HARD` scenario and shows comparable performance in the `SuperHARD` scenario, which demonstrates the transferability of MA$^2$E.

To further verify the effectiveness of MA$^2$E across diverse environments, additional experiments are conducted in GRF. We apply MA$^2$E to VDN. As in Figure 9, integrating MA$^2$E significantly improves performance in three different scenarios. Consequently, the above results demonstrate the extensibility of MA$^2$E for various MARL algorithms and domains. We evaluated the performance of VDN, QPLEX, RIIT, and another policy-based algorithm, IPPO (de Witt et al., 2020) with MA$^2$E in diverse scenarios and the results are illustrated in Appendix C. MA$^2$E consistently improves the performance across different algorithms and tasks.

## 4.4 ABLATION STUDIES

To identify a suitable structure and hyper-parameters of MA$^2$E, we conduct ablation studies across various settings. The comparison is conducted in the `3s_vs_5z` scenario of SMAC. The experimental results for comparison are illustrated in Figure 10 and in Appendix D.

**Masking Strategy**: We compare two masking strategies: agent-based and random masking. Random masking removes a random portion of the input data, irrespective of the agents. The masking ratio is randomly chosen from 0.15, 0.35, 0.5, 0.75, and 0.95, with the masking point also being randomly determined. This ratio follows the suggestion from Liu et al. (2023a). Figure 10 (a) compares win rates based on different masking strategies and agent-based masking is superior because it can effectively capture correlations between agents.

**The Number of Trajectories**: To test the appropriate trajectory length, we conduct a comparison of the number of trajectories in five different settings : *1*, *3*, *5*, *7*, and *9*. Figure 10 (b) compares performance based on the number of trajectories. The performance does not show a consistent difference among the different number of trajectories. However, it can be observed that the performance is at its best when the number of trajectories is *5*, and it deteriorates as the number becomes smaller or larger than *5*. Therefore, to achieve good performance with MA$^2$E, it is necessary to set appropriate number of trajectories depending on the scenario or environment.

**Positional Encoding**: To test the effect of the positional encoding design, we compare positional encoding using three different encoding methods : *Agent*, *Time*, and *Both*. *Agent* encoding distinguishes encoding based on agents, while *Time* encoding sets different values based on the trajectories. In the *Both* setting, encoding takes both *Agent* and *Time* into account. For example, half of the embedding space is dedicated to agent-based encoding, and the remaining half is utilized for time-based encoding. Figure 10 (c) compares performance based on the positional encoding design. As we can see, the performance is significantly better when considering both time and agent, as opposed to either of them individually. When considering only one of either time or agent, there is not a significant difference in performance compared to the case where MA$^2$E is not used. In particular, when considering only time, there are intervals where the performance is even worse than without MA$^2$E.

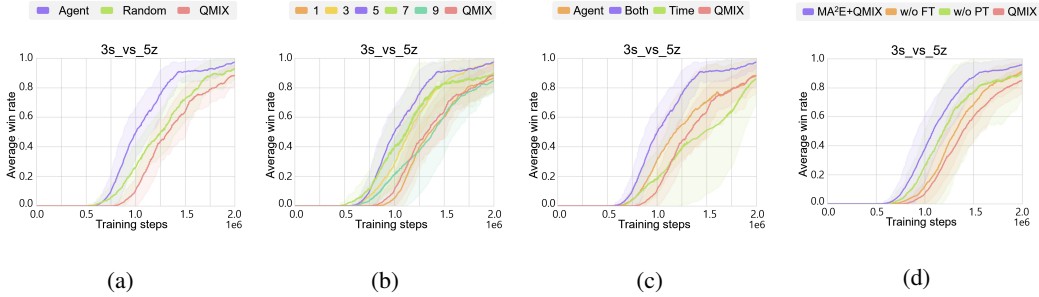

Figure 10: Performance comparison based on hyper parameter settings or strategies. **(a)**: Performance comparison based on masking strategies. The agent-based masking strategy applied model outperforms both the random masking strategy applied model and the default model. **(b)**: Performance comparison of MA$^2$E based on the number of trajectories. **(c)**: Performance comparison of MA$^2$E based on the positional encoding strategies. When considering both agent and time, it shows the best performance. **(d)**: Performance comparison of MA$^2$E with or without pre-training and fine-tuning.

**MA$^2$E Training**: As mentioned in Section 3, MA$^2$E training is divided into two stages. To verify the importance of both stages, we report the performance of MA$^2$E without pre-training (`w/o PT`) and without fine-tuning (`w/o FT`). As shown in Figure 10 (d), employing both pre-training and fine-tuning achieves the best performance. Moreover, there is a tendency for the performance of the case without pre-training (`w/o PT`) to be better than the case with pre-training only (`w/o FT`), but the variance of `w/o PT` is very high. This indicates that when MA$^2$E is underfitted and its outputs used in decision making are likely to act as noise. Therefore, pre-training is necessary to reliably improve the performance and fine-tuning is required to align MA$^2$E with the policy.

## 5 CONCLUSION AND LIMITATIONS

In this paper, we propose a novel method MA$^2$E, which utilizes the masked auto-encoder to address partial observability in the context of multi-agent reinforcement learning. MA$^2$E empowers agents to derive global insights solely from own local information, thereby enhancing their collaborative decision-making capabilities. The proposed method seamlessly integrates with both value-based and policy-based MARL algorithms. Through extensive experimentation, we have substantiated its effectiveness in enhancing sample efficiency and elevating task-solving proficiency across a diverse set of scenarios within the SMAC, SMACv2, and GRF environments. We posit that the incorporation of MA$^2$E into MARL, which extends the observational horizon of agents, stands as a pivotal advancement in MARL research. By introducing a methodology that addresses the challenge of partial observability in MARL, MA$^2$E has emerged as a cornerstone in the field.

Nonetheless, our proposed model exhibits limitations. When agents are positioned far apart such that there is no overlap in their observations, the capacity to infer information about other agents is severely restricted, potentially resulting in the inference outcomes of MA$^2$E being not useful. A more comprehensive delineation of these limitations can be found in Appendix H. Additionally, the structural characteristics of MA$^2$E make it relatively difficult to scale to a varying number of agents. In future work, we plan to investigate approaches that enable scalability, such as team composition.

ACKNOWLEDGEMENT

This work was supported by Institute of Information & communications Technology Planning & Evaluation (IITP) grant funded by the Korea government(MSIT) (No. RS-2024-00457882, AI Research Hub Project), by Institute for Information & communications Technology Planning & Evaluation(IITP) grant funded by the Korea government(MSIT) (RS-2019-II190075, Artificial Intelligence Graduate School Program(KAIST)), by Institute of Information & communications Technology Planning & Evaluation (IITP) under Open RAN Education and Training Program (IITP-2025-RS-2024-00429088) grant funded by the Korea government(MSIT), and conducted by

Center for Applied Research in Artificial Intelligence (CARAI) grant funded by DAPA and ADD (UD230017TD).

## REPRODUCIBILITY STATEMENT

For the details of environments and hyperparameters, please refer Section 4 and Appendix A. To run our method, please download the supplementary material and follow the instructions in README files. We employed pymarl2 (Hu et al., 2021) or the official codes from the authors for baselines.

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

# A  EXPERIMENTAL DETAILS

In this section, we introduce the environments used in the experiments, the baseline algorithms, as well as the hyperparameters and computational resources. Experiments are carried out on NVIDIA A6000 and GTX3090 GPUs and AMD EPYC 7313 CPU.

## A.1  ENVIRONMENTS

We conduct experiments in the following environments:

- StarCraft Multi-Agent Challenge (SMAC) (Samvelyan et al., 2019) from `https://github.com/oxwhirl/smac` which is licensed under MIT license.
- SMACv2 (Ellis et al., 2023) from `https://github.com/oxwhirl/smacv2` which is licensed under MIT license.
- Google Research Football (GRF) (Kurach et al., 2020) from `https://github.com/google-research/football` which is licensed under Apache License 2.0.

All algorithms are implemented based on the open-source framework *pymarl2* (Hu et al., 2021) from `https://github.com/hijkzzz/pymarl2` which is an augmented version of *pymarl* from `https://github.com/oxwhirl/pymarl`. Both are licensed under Apache License 2.0.

### A.1.1  SMAC

The StarCraft Multi-Agent Challenge (SMAC) is one of the benchmarks widely utilized in research to evaluate MARL algorithms. Units from the strategy video game StarCraft II engage in confrontations with each other in diver scenarios. The objective is for multiple agents to collaborate in defeating the enemies. There are multiple scenarios, each categorized into difficulty levels such as `EASY`, `HARD`, and `SuperHARD`. We primarily conduct experiments in `HARD`, and `SuperHARD` scenarios. Table 3 provides a detailed description of the scenarios we used in our experiments.

| Scenario | Difficulty | Ally Units | Enemy Units | Type |
|---|---|---|---|---|
| `2s_vs_1sc` | EASY | 2 Stalkers | 1 Spine Crawler | micro-trick: alternating fire |
| `3s_vs_3z` | EASY | 3 Stalkers | 3 Zealots | micro-trick: kiting |
| `3s_vs_5z` | HARD | 3 Stalkers | 5 Zealots | micro-trick: kiting |
| `2c_vs_64zg` | HARD | 2 Colossi | 64 Zerglings | micro-trick: positioning |
| `MMM` | HARD | 1 Medivac 2 Marauders 7 Marines | 1 Medivac 2 Marauders 7 Marines | heterogeneous & symmetric |
| `corridor` | Super HARD | 6 Zealots | 24 Zerglings | micro-trick: wall off |
| `6h_vs_8z` | Super HARD | 6 Hydras | 8 Zealots | micro-trick: focus fire |
| `MMM2` | Super HARD | 1 Medivac 2 Marauders 7 Marines | 1 Medivac 3 Marauders 8 Marines | heterogeneous & asymmetric |
| `1o_2r_vs_4r` | - | 1 Overload 2 Roaches | 4 Roaches | communication |

Table 3: A detailed description of the SMAC scenario used in the experiment

### A.1.2  SMACv2

SMACv2 is proposed to address the shortcomings of SMAC, particularly in terms of its lack of stochasticity and partial observable characteristics (Ellis et al., 2023). Therefore, SMACv2 differs from SMAC in three main aspects.

First, the unit composition is randomly determined. In SMAC, the generated units are fixed, whereas in SMACv2, different types of units are randomly generated based on probabilities. The second difference lies in the observation probability of agents. In SMAC, when one agent observes an enemy, other agents can also observe the same enemy simultaneously. In contrast, in SMACv2, if one agent observes an enemy first, other agents within their observation range may not identify the same enemy, even if it is present. The last distinction involves adding randomness to the location where units are spawned. The location where units are spawned is determined by one of two types: one is *surround* and the other is *reflect. surround* entails the creation of units in a formation where allied units surround enemy units, while *reflect* involves units being spawned in a facing and confronting manner.

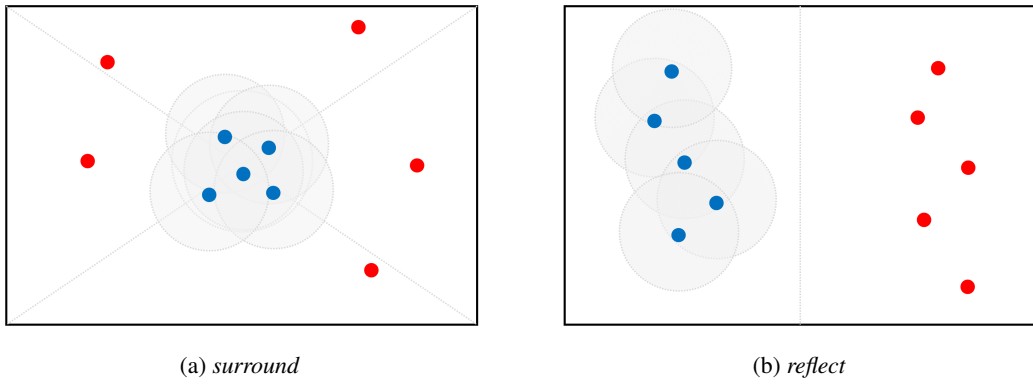

(a) *surround*                    (b) *reflect*

Figure 11: Two different types of start positions in SMACv2

We conduct experiments in SMACv2 under conditions that encompass all three aforementioned aspects, and additionally, we fixed the last condition to investigate performance differences in MA$^2$E based on the spawning locations. To elaborate further, when comparing the results between the *surround* and *reflect*, we assumed that *surround* , where agent observations already sufficiently overlap, would render global information less significant. On the other hand, in the case of *reflect*, agents are spawned at an appropriate distance, allowing for the assumption that through the inference of global information, they could acquire useful information about agents at long range.

Figure 11 provides examples of the *reflect* and *surround* positions. The light gray circles represent the observation range of allied agents. In the case of *surround*, most areas overlap. In contrast, *reflect* shows a relatively smaller overlap compared to *surround*. Therefore, we believe that MA$^2$E can better infer global information in the *reflect* position than in the *surround* position.

### A.1.3 GOOGLE RESEARCH FOOTBALL (GRF)

Google Research Football (GRF) is one of the widely used benchmark in multi-agent reinforcement learning research. Multiple agents cooperate to play a football game in GRF and GRF offers various scenarios. We experiment with our model in three different scenarios of GRF.

- *academy_3_vs_1with_keeper*: The objective is for three allied agents to score goals in a soccer half-court field against an opposing goalkeeper.
- *academy_counterattack_easy*: The scenario consists of four allied agents, one opposing agent, and one goalkeeper. The objective is for the allied agents to execute a counter-attack.
- *academy_counterattack_hard*: The scenario consists of four allied agents, two opposing agents, and one goalkeeper. The objective is for the allied agents to execute a counter-attack.

### A.2 HYPERPARAMETERS

The hyperparameters for the baselines in our experiments are as listed in Table 4. The value-based algorithms include VDN, QMIX, QPLEX, ow-QMIX, cw-QMIX, while the policy-based algorithms include IPPO, RIIT. Moreover, the hyperparameters for MA$^2$E are listed in Table 5.

Table 4: The hyperparameter settings for the baseline algorithms

| Algorithms | Value-based | Policy-based |
|---|---|---|
| Optimizer | Adam | Adam |
| Batch Size | 128 | 64, 32 |
| TD($\lambda$) | 0.6 | - |
| Learning Rates | 0.001 | 0.0005, 0.001 |
| Replay Buffer Size | 1000 | 64, 128 |
| $\epsilon$ Anneal Steps | 100000 | 100000 |
| Gamma | 0.99 | |

Table 5: The hyperparameter settings for MA$^2$E

| Hyperparameters | Value |
|---|---|
| Batch size | 32 |
| Input embedding | 24 |
| The number of heads | 4 |
| The number of encoder layer | 3 |
| The number of decoder layer | 2 |
| Steps for fine tuning | 500 |
| Pretraining threshold | 0.015 |

# B PSEUDOCODE

---

**Algorithm 1** Model with Multi-Agent Masked Auto-Encoder (MA$^2$E) Applied

---

1: Initialized value networks $Q_{\theta_{ind}}$ and $Q_{\theta_{tot}}$ or policy network $\pi_\theta$
2: Initialized MA$^2$E parameters $\theta_{ma^2e}$
3: Prepare replay buffer $D$
4: **repeat**
5:     Run episodes through random policy and store trajectories in Buffer $D$
6:     Update MA$^2$E parameters $\theta_{ma^2e}$ using samples in $D$ : $L_{MA^2E} = \frac{1}{nT} \sum_{t=1}^{T} \sum_{i=1}^{n} (\tau_t^i - \widetilde{\tau}_t^i)^2$
7: **until** When training has been done for a specific number of steps or when the loss is lower than the threshold
8: Reset replay buffer $D$
9: **repeat**
10:     **for** *each episode* **do**
11:         Get initial state $s$
12:         **while** *episode* is not terminated **do**
13:             Sample actions $\mathbf{a}_t$ from $Q$ with $\epsilon$ greedy or policy $\pi_\theta$
14:             Execute actions and observe reward $r_t$
15:             Store transition $(s_t, \mathbf{o}_t, \mathbf{u}_t, r_t)$ in buffer $D$
16:         **end while**
17:         Update value networks $Q_{\theta_{ind}}$ and $Q_{\theta_{tot}}$ or policy network $\pi_\theta$
18:         Update MA$^2$E parameters $\theta_{ma^2e}$
19:     **end for**
20: **until** reaching maximum total environment steps

---

## C ADDITIONAL RESULTS OF MA²E WITH VDN, QPLEX, RIIT AND IPPO ACROSS DIFFERENT ENVIRONMENT

We provide additional experimental results from SMAC, SMAC2, and GRF. All results were tested with 3 random seeds.

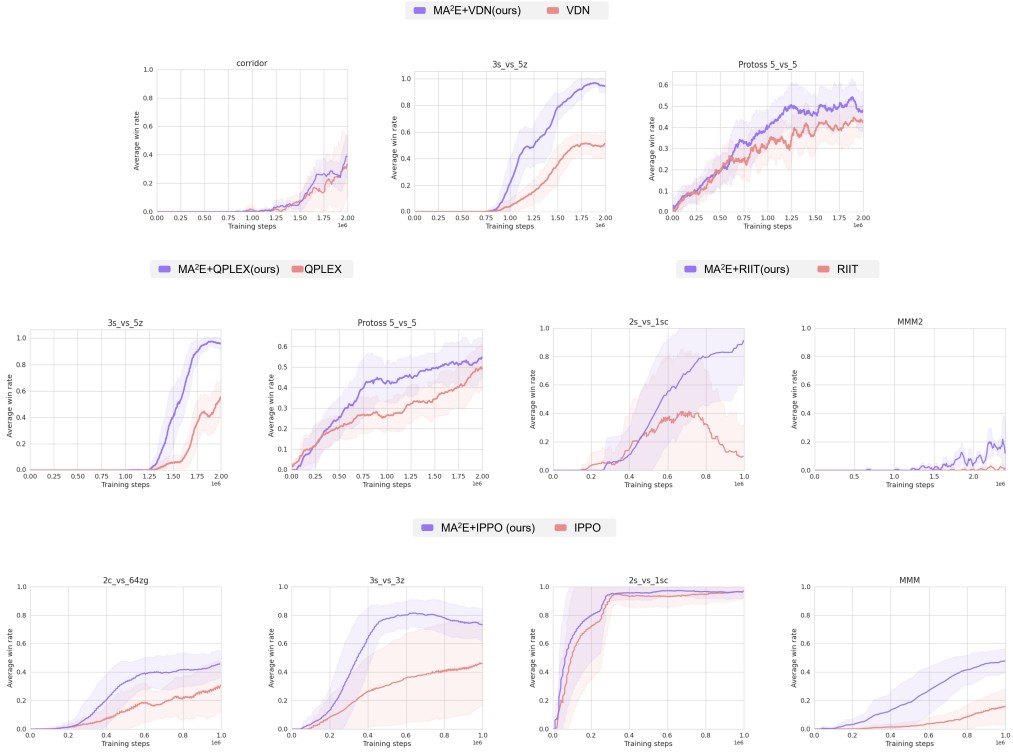

Figure 12: Performance comparison in SMAC and SMACv2 scenarios

## D ADDITIONAL ABLATION STUDY

We provide additional ablation study in 2c_vs_64zg, similar to Figure 10. We can draw a similar conclusion that: using agent-level masking, an appropriate number of trajectories, positional encoding considering both agent and time, and both pretraining and fine-tuning yield better results.

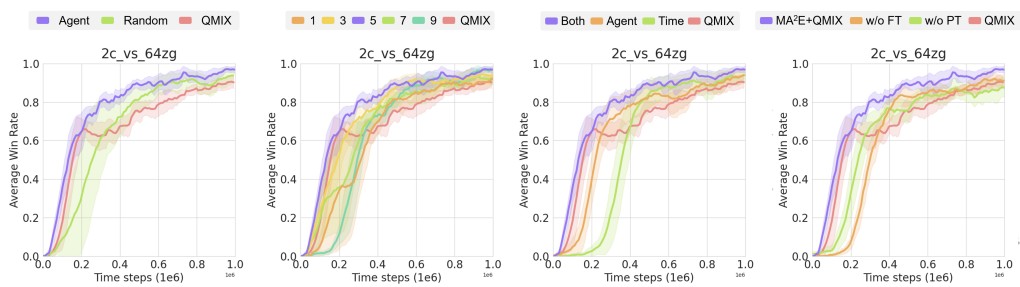

Figure 13: Additional ablation study in 2c_vs_64zg scenario. Along with Figure 10, we can draw a similar conclusion.

# E   LEARNING CURVE OF TABLE 2

We provide learning curves of each algorithm in Table 2. As shown in Figure 14, QMIX+MA$^2$E outperforms the baselines in most scenarios.

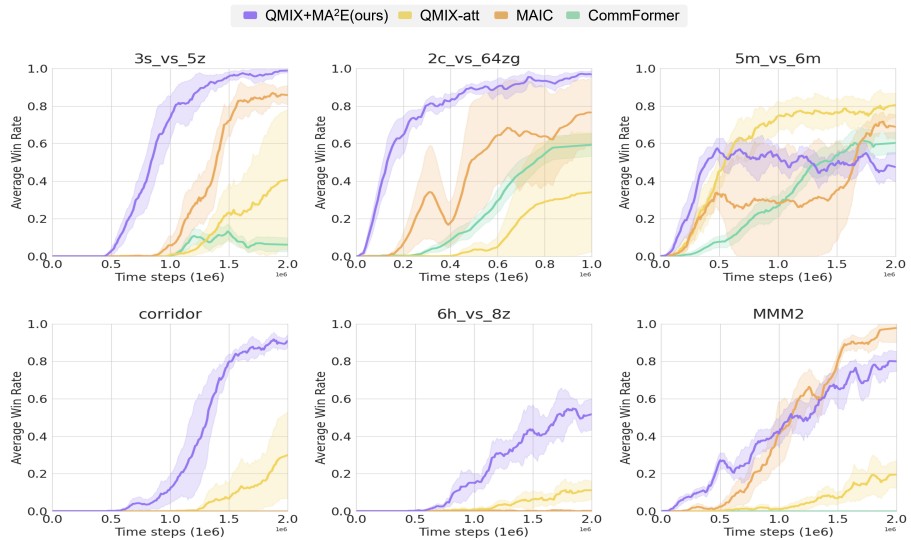

Figure 14: Learning curves of each algorithm in Table 2.

# F   COMPARISON OF TRAINING COST AND MODEL SIZE WITH AND WITHOUT USING MA$^2$E

| 3s_vs_5z | MA2E+QMIX | QMIX | Difference |
|---|---|---|---|
| Time for pretraining | 0.77h | - | +0.77h |
| Time to convergence | 17.05h (faster) | 18.22h | -1.17h (93%) |
| Steps to convergence | 2.03M (faster) | 3.52M | -1.49M (57%) |
| Steps per second | 33.07 | 53.66 | -20.59 |
| The number of parameters | 0.43M | 0.12M | 0.31M |

Table 6: Comparison of training and execution time and the number of parameters for each method in the SMAC 3s_vs_5z scenario. The values in parenthesis represent the relative ratio between MA2E+QMIX and QMIX.

| 2c_vs_64zg | MA2E+QMIX | QMIX | Difference |
|---|---|---|---|
| Time for pretraining | 0.84h | - | +0.84h |
| Time to convergence | 7.97h (faster) | 8.76h | -0.79h (90%) |
| Steps to convergence | 0.639M (faster) | 1.021M | -0.382M (63%) |
| Steps per second | 22 | 32 | -10 |
| The number of parameters | 0.98M | 0.65M | 0.33M |

Table 7: Comparison of training and execution time and the number of parameters for each method in the SMAC 2c_vs_64zg scenario. The values in parenthesis represent the relative ratio between MA2E+QMIX and QMIX.

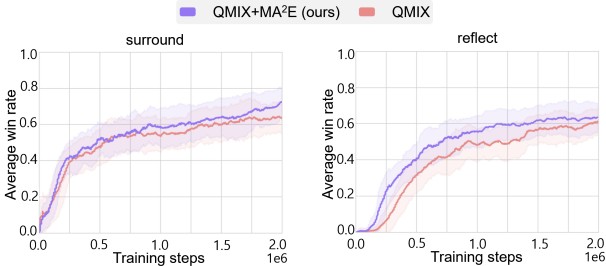

Figure 15: Performance comparison according to the different starting position in SMACv2

## G    ABLATION STUDIES ON OBSERVATION OVERLAP

SMACv2 offers different starting positions based on the configurations, with two main strategies for positioning: *surround*, where the units are placed in a position surrounded by the enemy, and *reflect*, where they are positioned head-on in a confrontation with the enemy. The extent of overlap in agent observations varies depending on each position. (For more details, please refer to Appendix A.1.2.) We utilize these two positions to compare the performance of MA$^2$E based on the extent of observation overlap. In both cases, the performance is better than the baseline, but it is particularly evident from Figure 15 that the performance is significantly better in the *reflect* compared to the baseline. Therefore, it can be concluded that when the observation range overlaps at a reasonable proportion, MA$^2$E can better infer global information, as opposed to cases where the overlap is too extensive.

## H    LIMITATIONS

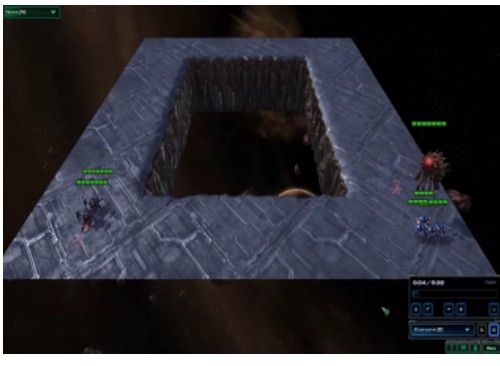

(a) A Screenshot of `1o_2r_vs_4r` scenario in SMAC

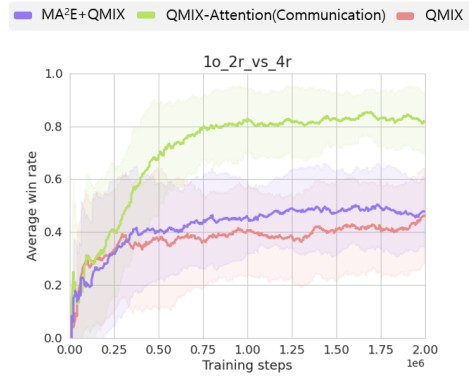

(b) Win rate comparison between using communication and using MA$^2$E.

Figure 16: A scenario where MA$^2$E fails to infer global information

An agent must use observations as clues to infer the actions or movements of other agents. Therefore, if there is no information about other agents in the observation of an agent, MA$^2$E cannot effectively capture the information.

For example, in environments where agents are spread far apart due to a large map size, and their observations do not overlap, MA$^2$E may struggle to infer the information. Figure 16a shows a screenshot of the `1o_2r_vs_4r` scenario, and Figure 16b shows a comparing the win rates when the model uses communication and when the model uses MA$^2$E. In this map, enemies and allies spawn at random locations among the four corners (top-left, bottom-left, top-right, bottom-right) of

the map. Allied agents consist of two attacking units and one observation unit, with the observation unit spawning at the same location as the enemies.

If communication is possible, the observation unit can inform the attacking units of the enemy's location. On the other hand, since the observation unit is outside the observation range of the attacking units, they cannot infer information about the observation unit. We compare our proposed model with the communication method presented by Hu et al. (2021), called QMIX-Attention, to investigate this situation. As shown in the Figure 16b, the performance of our proposed model is relatively lower compared to the case where communication is used. Nevertheless, despite that, it can be confirmed that using MA$^2$E does not make lower performance compared to the default model.

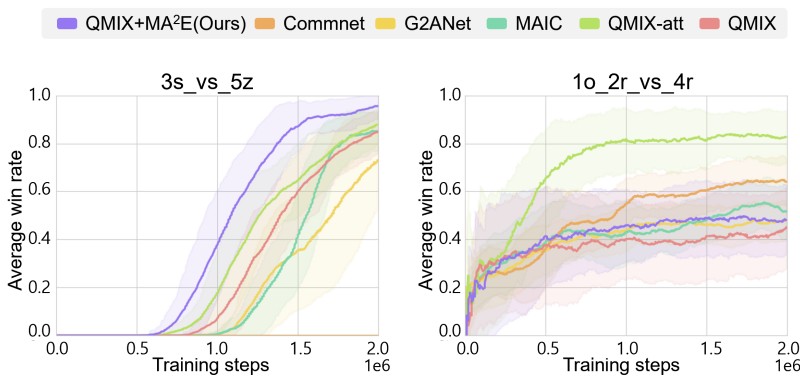

Figure 17: Performance comparison with communication based MARL algorithms in two SMAC scenarios

We also compare with communication methods Commnet (Sukhbaatar et al., 2016), G2ANet (Liu et al., 2020), MAIC (Yuan et al., 2022), and QMIX-att (Hu et al., 2021). The results are shown in Figure 17. In `3s_vs_5z`, appropriately overlapping observations allow MA$^2$E to effectively infer information, and the use of attention layers reduces the space size. Conversely, communication based methods increase the space size and require communication overhead, leading to slower convergence and sample inefficiency. However, in `1o_2r_vs_4r`, MA$^2$E struggles to perform effectively due to a very little observation overlap among agents, while communication methods (QMIX-att and Commnet) can still fully share the information hence achieves better performance. Thus, we note that the need for communication vary depending on the environment.

## I   IS MA$^2$E PERFORMING WELL BECAUSE IT IS OVERFITTED TO THE ENVIRONMENT?

MA$^2$E learns the dynamics of the environment during the training process and captures global information through its ability to infer masked areas. Therefore, from one perspective, it can be considered that MA$^2$E performs well due to overfitting to the environment. Specifically, in the SMAC environment, there is a significant performance difference between algorithms applying MA$^2$E and the baseline, but in the SMACv2 environment, this difference is not as pronounced. This suggests that MA$^2$E may be overfitting to the environment because SMAC is more deterministic compared to SMACv2, making it easier for the model to overfit.

However, the issue of not performing well in SMACv2 is a common difficulty not only for MA$^2$E but also for existing MARL algorithms. In reflecting the stochastic characteristic of SMACv2, the random properties can generate scenarios that are unwinnable from the start or turn each episode into a completely different task, requiring a multi-task approach. Therefore, it is difficult to argue that MA$^2$E is overfitting just because it does not perform well in SMACv2.

Nonetheless, to prove that MA$^2$E is not overfitting to the environment, we created a non-deterministic scenario by adding random characteristics to the `3s_vs_5z` scenario of SMAC and used it for

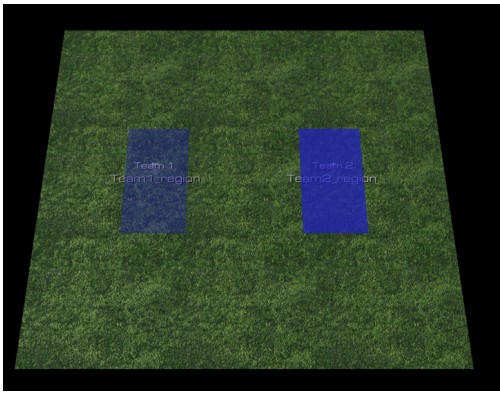
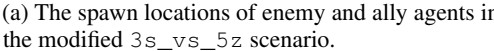

(a) The spawn locations of enemy and ally agents in the modified `3s_vs_5z` scenario.

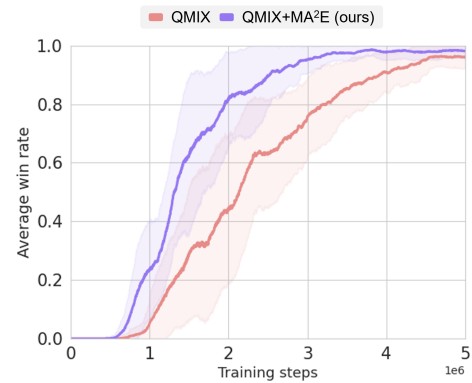

(b) Win rate comparison between using MA$^2$E and without MA$^2$E case

Figure 18: The experimental results in the environment with added stochastic characteristics

experimentation. In the original `3s_vs_5z` scenario, the positions where enemy and ally agents spawn are fixed. In contrast, in our modified scenario, the enemy and ally agents spawn at random locations within the blue area shown on the Figure 18a. Thus, a stochastic property is added where the spawn positions of the enemy and ally agents change in each episode.

The experimental results are shown on the Figure 18b. As a result, it can be seen that the model using MA$^2$E significantly outperforms the model without it. Therefore, it can be concluded that MA$^2$E does not overfit to the environment but appropriately infers global information according to the situation.

## J  BROADER IMPACT

Our study introduces Multi-Agent Masked Auto-Encoder (MA$^2$E) to enhance decision-making in multi-agent systems. By addressing partial observability through a masking perspective, our approach has broad applications in real-world scenarios such as military operations, autonomous driving, traffic systems, and robotics, promising improved decision-making across diverse multi-agent environments. While explicit communication can selectively send the messages, MA$^2$E is trained with full information. Hence the proposed method may cause privacy or security issues in some real world applications, which would be mitigated by anonymizing data.

