# OpenReview forum: "MA$^2$E: Addressing Partial Observability in Multi-Agent Reinforcement Learning with Masked Auto-Encoder"
_ICLR.cc/2025/Conference — ICLR 2025 Poster_

### Official Review · Reviewer_2kwB · 2024-10-25

**Soundness:** 3
**Presentation:** 3
**Contribution:** 2
**Rating:** 5
**Confidence:** 5

**Summary:**

This paper focuses on multi-agent reinforcement learning under partial observability and proposes to reconstruct the agents' trajectory information through a Multi-Agent Masked AutoEncoder (MA$^2$E). During execution, the agent inputs its trajectory into MA$^2$E to infer other agents' trajectories, which is then processed by a self-attention layer. Finally, the agent aggregates the information from its observation and the inference to produce the action. The authors conducted experiments on SMAC and GRF to assess the effect of MA$^2$E.

**Strengths:**

1. This paper is easy to follow. The motivation is clear, and the presentation is of good quality.
2. The authors conducted sufficient experiments, which provide a comprehensive demonstration of MA$^2$E.
3. The proposed method can be applied to various multi-agent reinforcement learning algorithms, including value-based and policy-based.

**Weaknesses:**

1. The novelty of this paper is relatively limited. It simply integrates a masked autoencoder into the decision-making process of agents, which is more oriented to engineering and makes little theoretical contribution. It
2. Some ablation studies are conducted on only a single scenario (e.g., 3s_vs_5z). The results may be contingent.
3. The selected baselines are not strong enough. Many MARL algorithms also modify the decision-making process of agents while keeping the constraints in observability and communication, such as RODE [1], LDSA [2], and SORA [3]. The authors should compare MA$^2$E with these methods to make the results more convincing.
4. Compared to the baselines, the training of MA$^2$E is more complex and computationally expensive, but the performance is not significant enough. In other words, this method is not cost-effective.

[1] Wang, T.; Gupta, T.; Mahajan, A.; Peng, B.; Whiteson, S.; and Zhang, C. 2020c. Rode: Learning roles to decompose multi-agent tasks. arXiv preprint arXiv:2010.01523.
[2] Yang, M.; Zhao, J.; Hu, X.; Zhou, W.; Zhu, J.; and Li, H. 2022. Ldsa: Learning dynamic subtask assignment in cooperative multi-agent reinforcement learning. Advances in Neural Information Processing Systems, 35: 1698–1710.
[3] Zhou G, Xu Z, Zhang Z, et al. SORA: Improving Multi-agent Cooperation with a Soft Role Assignment Mechanism. International Conference on Neural Information Processing. Singapore: Springer Nature Singapore, 2023: 319-331.

**Questions:**

1. The official code of COLA is implemented based on the original *pymarl*, while MA$^2$E is implemented based on *pymarl2*. Is there any issue of fairness?
2. How does the training cost (measured in GPU hours) of MA$^2$E compare to that of other methods?

---

> ### Author Response · Authors · 2024-11-21
>
> Dear reviewer, thank you for your careful review and constructive comments. We greatly appreciate the opportunity to address your feedback. We present our responses below.
>
> ---
>
> **Weakness 1: The novelty of this paper is relatively limited. It simply integrates a masked autoencoder into the decision-making process of agents, which is more oriented to engineering and makes little theoretical contribution.**
>
> Our work introduces several novel contributions to the MARL domain, which are outlined as follows:
>
> - **Application of Masked AutoEncoder (MAE) in MARL**: We extended the principle of the Masked AutoEncoder, commonly used in the vision domain to infer masked areas from visible information, to address the partial observability problem in the MARL domain by inferring global information from local observations. This idea has been scarcely explored in the MARL domain, making it a novel application.
>
> - **Structural Considerations for MARL:** To directly apply the Masked AutoEncoder to MARL algorithms, it is necessary to modify its structure considering the characteristics of MARL. We proposed and validated an appropriate structure experimentally. Specifically, we applied masking at the agent level, designed positional encoding from the perspectives of time and agents, and leveraged historical information.
> - **Integration of MA²E into agents:** We proposed a method to incorporate MA²E into individual agents where each agent only receives its local information and selects actions. Furthermore, we employed the Masked AutoEncoder as a separate plug-in module, allowing MA²E to be easily integrated with existing MARL algorithms without modifying their core architecture.
> - **Extensive Experimental Validation:** The effectiveness of our contributions was demonstrated in complex MARL environments, including SMAC, SMACv2, and GRF, highlighting the broad applicability and practical impact of our approach.
>
> We also believe that theoretical insights would strengthen our paper. However, as is often the case in prior literature, achieving theoretical guarantees is inherently challenging. This difficulty is further amplified in our method, as it requires consideration of the characteristics of both MARL and Masked AutoEncoder. To address this, we conducted extensive experiments to validate effectiveness of our approach.
>
> ---
>
> **Weakness 2: Some ablation studies are conducted on only a single scenario (e.g., 3s_vs_5z). The results may be contingent.**
>
> Please refer to Figure 21. We conducted additional ablation studies in 2c_vs_64zg scenario. Together with the results in Figure 10, we can draw consistent conclusions: employing agent-level masking, using an appropriate number of trajectories, applying positional encoding that considers both agents and time, and utilizing both pretraining and fine-tuning lead to improved performance.
>
> ---
>
> **Weakness 3: The selected baselines are not strong enough. Many MARL algorithms also modify the decision-making process of agents while keeping the constraints in observability and communication, such as RODE, LDSA, and SORA. The authors should compare MA²E with these methods to make the results more convincing.**
>
> Thank you for suggesting strong baselines. We agree that comparing MA²E with these methods could make our results more convincing.
> Please refer to Figure 17 in the paper. We conducted additional experiments with RODE and LDSA. Unfortunately, we could not include SORA, as its code is not publicly available. Additionally, CDS, a strong baseline suggested by Reviewer dk6X, has been also included.
> As shown in Figure 17, RODE and LDSA fails to train in several scenarios. In contrast, MA²E consistently learns and improves performance across all scenarios. Furthermore, as illustrated in Table 6, QMIX+MA²E demonstrates the best average performance. These results further validate the effectiveness and robustness of our method.

---

> ### Author Response · Authors · 2024-11-21
>
> **Weakness 4: Compared to the baselines, the training of MA²E is more complex and computationally expensive, but the performance is not significant enough. In other words, this method is not cost-effective.**
>
> Please refer to Table 8 and Table 9 in the paper, where we analyze the training costs of MA²E+QMIX compared to the backbone QMIX method in two scenarios. As shown in the tables, the pre-training times required for MA²E are relatively small, accounting for only 4.5% and 10.5% of the total time to convergence, respectively.
> While applying MA²E to QMIX introduces a pre-training stage, MA²E+QMIX converges faster than QMIX, demonstrating improved sample efficiency and accelerated convergence in terms of both time and steps. This highlights the cost-effectiveness of the method. Moreover, our experimental results in the paper show that this enhanced sample efficiency reduces overall training time while achieving improved performance.
> We also report the measured pre-training times below:
>
> Scenario | 5m_vs_6m | 2c_vs_64zg | 3s_vs_5z | corridor | 6h_vs_8z | MMM2
> --- | --- | --- | --- | --- | --- | ---
> Pre-training time | 0.65h | 0.84h | 0.86h | 1.10h | 1.02h | 0.93h
>
> ---
>
> **Question 1: The official code of COLA is implemented based on the original pymarl, while MA²E is implemented based on pymarl2. Is there any issue of fairness?**
>
> The main difference lies in the StarCraft II verions used: the official COLA code and paper used SC2.4.6.2.69232, while our code used SC2.4.10. As noted in the official [SMAC](https://github.com/oxwhirl/smac) and [pymarl](https://github.com/oxwhirl/pymarl) repositories, performance is not always comparable between different versions. To ensure fairness, we used SC2.4.10 across all experiments including COLA. Furthermore, we carefully checked and followed the hyperparameters and settings in the paper.
>
> ---
>
> **Question 2: How does the training cost (measured in GPU hours) of MA²E compare to that of other methods?**
>
> We would greatly appreciate it if the reviewer could refer to Tables 8 and 9, as well as our response to Weakness 4, where we presented and discussed the training costs.
>
> ---
> We hope our responses have addressed your concerns. If there are any remaining issues, we would be grateful if you could discuss them with us during the discussion period. Thank you once again for your time and invaluable input.
>
> [1] Hu, Jian, et al. "Rethinking the Implementation Tricks and Monotonicity Constraint in Cooperative Multi-agent Reinforcement Learning." The Second Blogpost Track at ICLR 2023.

---

> > ### Comment · Reviewer_2kwB · 2024-11-22
> >
> > I really appreciate the authors' efforts in this work. After reading the comments from other reviewers and the rebuttals, I still have some concerns, and I list them below:
> >
> > - **The methodology and novelty**
> >
> > It is good that MA$^2$E uses the masked autoencoder to realize the reconstruction of global states while keeping the agents distributed. However, the good performance of MA$^2$E can be due to its well-designed decision-making process **or** the superiority of Transformer over MLP and the additional parameters. Therefore, I suggest the authors conduct further theoretical analysis or experiments on this issue.
> >
> > Besides, techniques such as Autoencoders, Masking, and Transformers have been widely introduced into MARL. The motivation of MA$^2$E also follows previous works like COLA. That's why I think the novelty is **relatively** limited, and therefore more significant experiment results are required to demonstrate the value of the proposed method.
> >
> > - **Rigor in experiments**
> >
> > The results of MA$^2$E shown in the paper are very significant. However, according to the submitted code and line 530 in the updated draft, MA$^2$E is implemented based on *pymarl2*, while other baselines are implemented with their official codes. An important issue is that MA$^2$E and the baselines employ different hyperparameter settings. For example, the batch size and learning rate in COLA are set to 32 and 0.0005 respectively, while 128 and 0.001 in MA$^2$E. Such an issue also exists in the RODE and LDSA results in Figure 17. As we know, larger batch sizes and higher learning rates can speed up model learning and improve the performance of MA$^2$E. Besides, the settings of other hyperparameters in MA$^2$E, such as optimizer and intensity of exploration, are also different from the baselines implemented based on *pymarl*. Therefore, the comparison between MA$^2$E and the baselines is unfair, leading to unconvincing results. I would like the authors to provide further explanation for this.

---

> > > ### Author Response · Authors · 2024-11-22
> > >
> > > Dear reviewer, thank you for your detailed and insightful feedback. We present our responses below.
> > >
> > > ---
> > >
> > > **The good performance of MA²E can be due to the superiority of Transformer over MLP and additional parameters.**
> > >
> > > Thank you for your invaluable suggestion. We believe that comparing our method with a baseline that adopts a Transformer architecture and additional parameters would provide valuable insights and further enhance our work. To validate the effectiveness of our method more comprehensively, we plan to conduct additional experiments using such a baseline. We will update the results as soon as possible to address your concern.
> > >
> > > ---
> > >
> > > **Relatively limited novelty**
> > >
> > > Thank you for your insightful feedback. We believe that our contribution lies in effectively leveraging such techniques to address partial observability. To demonstrate the effectiveness of our method, we compared it with a baseline that employs reconstruction and masking principles (MA2CL). Additionally, we will update our comparison to include a baseline adopting Transformer-based architectures as soon as possible.
> > >
> > > While the motivations behind MA²E and COLA may share similarities, we believe that novelty can also arise from differences in approach and performance. COLA infers the same one-hot consensus for all agents from different local observations. As COLA uses a fixed number ($K$) of consensus representations for decision-making, its expressive capability may be limited in complex tasks. As noted in the COLA paper, the choice of K depends on the difficulty of the task, and it requires additional effort to determine an appropriate K value. In contrast, MA²E infers masked (unseen, global) areas from unmasked (local) observations by leveraging the principles of a Masked Auto-Encoder. The effectiveness of our approach is demonstrated in experiments.
> > > As noted above, we will conduct additional experiments using the same hyperparameter settings to further validate the value of MA²E.
> > >
> > > ---
> > >
> > > **Rigor in experiments**
> > >
> > > Thank you for your valuable suggestion. We originally chose to use the hyperparameters provided in the official implementations of the baseline papers, as we assumed these were optimized for their respective methods and would ensure reliable and fair comparisons. Our intention was to respect the original settings used by the authors.
> > > Nevertheless, we agree with the reviewer’s suggestion that using the same hyperparameters across methods could further strengthen our comparisons.
> > > - For QMIX, VDN, OW-QMIX, CW-QMIX, and QPLEX, we confirm that all evaluations were conducted using the same hyperparameter settings as MA²E.
> > > - For the other baselines, we will conduct additional experiments using the same hyperparameter settings as MA²E, including a batch size of 128, a learning rate of 0.001, the same optimizer, exploration epsilon, and buffer size. We will update the results as soon as possible.
> > >
> > > ---
> > >
> > > Once again, we sincerely appreciate your valuable feedback. We will make every effort to address your concerns.

---

> > > > ### Comment · Reviewer_2kwB · 2024-11-23
> > > >
> > > > Thanks for your reply. The novelty of the paper is a relatively subjective judgement, and the rigor in experiments is a more important issue. If the experiment results are not convincing, then the superiority of MA$^2$E can not be demonstrated. As shown in [1], all the selected MARL methods can benefit from the tuned hyperparameters. So if the authors deliberately ignore differences in hyperparameter settings, they may be suspected of intentionally lowering the baselines. Therefore, I have to keep my score. MA$^2$E will be a good method if it is as superior as shown in this paper after conducting new experiments with the same tricks and hyperparameters (e.g., implementing all the algorithms with the same framework *pymarl* or *pymarl2* and keeping the default hyperparameter settings unchanged).
> > > >
> > > > [1] Hu, Jian, et al. "Rethinking the Implementation Tricks and Monotonicity Constraint in Cooperative Multi-agent Reinforcement Learning." The Second Blogpost Track at ICLR 2023.

---

> ### Author Response · Authors · 2024-11-25
>
> Dear reviewer, we appreciate your valuable comments. Below, we provide detailed responses to your remaining concerns.
>
> ---
>
> **Rigor in Experiments**
>
> To address your concerns regarding the rigor of our experiments, we conducted additional experiments. The results are presented in Figure 23 and Table 10. The experimental settings are specified as follows:
>
> - **Framework**
>
>     As the reviewer’s main concern was the alignment of the framework and hyperparameters, we utilized baselines implemented in pymarl or pymarl2 (e.g., MA2CL was not implemented in either pymarl or pymarl2). All experiments were conducted using pymarl2, which builds upon pymarl with additional MARL methods and tricks. This allowed the baselines to be easily integrated into the pymarl2 framework.
>
> - **Hyperparameter**
>     - For QMIX, VDN, OW-QMIX, CW-QMIX, and QPLEX, we ensured that all evaluations were conducted using the same hyperparameter settings as MA²E.
>     - For COLA, LDSA, RODE, the hyperparameters were aligned with those used for MA²E, as detailed below:
>
>     | **lr(learning rate)** | **batch size** | **optimizer** | **epsilon_start** | **epsilon_finish** | **epsilon_anneal_time** | **action_selector** | **test_nepisode** |
>     | --- | --- | --- | --- | --- | --- | --- | --- |
>     | 0.001 | 128 | Adam | 1.0 | 0.05 | 100000 | epsilon_greedy | 32 |
>
>     | **buffer_size** | **mixing_embed_dim** | **hypernet_embed** | **runner** | **td_lambda (TD(λ))** | **gamma** | **batch_size_run** | **rnn_hidden_dim** |
>     | --- | --- | --- | --- | --- | --- | --- | --- |
>     | 1000 | 32 | 64 | parallel | 0.6 | 0.99 | 8 | 128 |
>
>     - RODE: As specified in the original paper, `n_role_clusters` was set 5 for MMM2 and 3 for other scenarios.
>     - COLA: The original paper indicates values of `K = 4` or `K = 32`. We trained models using both values for each scenario and report the best results.
> - As shown in Figure 23 and Table 10, MA²E consistently outperforms other methods, even when using the same hyperparameter settings. These results show the superiority of MA²E and demonstrate the rigor of our experiments.
>
> ---
>
> **Comparison with a Transformer-based Method**
>
> We conducted additional experiments with a baseline that adopts a transformer architecture, TransMix [1], to investigate whether the good performance of MA²E is due to the transformer and additional parameters. The experiments were carried out under the same settings as MA²E and pymarl2.
>
> As shown in Figure 23 and Table 10, despite both methods utilizing Transformer architectures, MA²E consistently outperforms TransMix. This suggests that the superior performance of MA²E stems not merely from the use of the transformer but also from how effectively it is leveraged. We believe that the novelty could lie in how effectively such techniques are utilized.
>
> ---
>
> Once again, thank you for your prompt and thoughtful feedback.
>
> [1] Khan, M. J., Ahmed, S. H., and Sukthankar, G. TransformerBased Value Function Decomposition for Cooperative Multi-Agent Reinforcement Learning in StarCraft. Proceedings of the AAAI Conference on Artificial Intelligence and Interactive Digital Entertainment, 18(1):113–119, Oct. 2022.

---

> > ### Comment · Reviewer_2kwB · 2024-11-25
> >
> > I appreciate the authors' time and efforts. I've checked the additional experiment results, and here's my comments:
> >
> > - The new experiments with aligned hyperparameter settings surely make the study more rigorous. A minor issue is that baselines such as COLA and RODE have specific modules like the masked autoencoder in MA$^2$E. The hyperparameters for these modules are tuned based on the episode runner. When the runner is changed to parallel, the number of iterations is also changed and the old hyperparameters may no longer fit in. I understand that due to time constraints, it is not practical to re-experiment with episode runner. Therefore, I won't require additional experiments and would like to raise the score.
> >
> > - The TransMix is not a suitable baseline for the ablation study, as it employs Transformer in the mixing network while MA$^2$E uses Transformer in the agent network. Since it is hard to redesign a new Transformer-based agent network, it seems more feasible to prove theoretically that the reconstruction process benefits the agent's decision-making.

---

> > > ### Author Response · Authors · 2024-11-28
> > >
> > > Dear reviewer, we sincerely appreciate your invaluable feedback.
> > >
> > > To address your final concern regarding Transformer, we conducted additional experiments with ATM [1] and TransfQMIX [2], both of which adopt the Transformer in their agent networks. The results are presented in Figure 24. These experiments were conducted under the same settings as MA²E. Although all methods utilize the Transformer in the agent network, MA²E shows superior performance compared to the other baselines.
> > >
> > > Furthermore, we conducted additional experiments using MA²E without masking and its associated loss while retaining the Transformer architecture in the agent network. The results are labeled as QMIX+Transformer to indicate MA²E without masking and are presented in Figure 25. These experiments were also conducted under the same settings as MA²E. Despite incorporating the Transformer architecture in the agent network, the performance was lower than that of the original MA²E, especially in super-hard corridor` and 6h_vs_8z scenarios.
> > >
> > > These results suggest that the superiority of MA²E arises not only from the use of the Transformer but also from its effective integration with masking strategies, thereby highlighting the novelty of our approach.
> > >
> > > Once again, we sincerely appreciate your time and effort in reviewing our work.
> > >
> > > ---
> > >
> > > [1] Yang, Yaodong, et al. "Transformer-based working memory for multiagent reinforcement learning with action parsing." *Advances in Neural Information Processing Systems* 35 (2022): 34874-34886.
> > >
> > > [2] Gallici, Matteo, Mario Martin, and Ivan Masmitja. "TransfQMix: Transformers for Leveraging the Graph Structure of Multi-Agent Reinforcement Learning Problems." *Proceedings of the 2023 International Conference on Autonomous Agents and Multiagent Systems*. 2023.

---

> ### Author Response · Authors · 2024-12-03
> **Kind Reminder: Discussion Deadline Approaching**
>
> Dear Reviewer 2kwB,
>
> It has been our pleasure to receive your valuable feedback throughout this discussion. As the discussion period nears its conclusion, we would like to kindly remind you that we have provided updates in our previous comment addressing your final concern regarding the Transformer architecture.
>
> We would greatly appreciate it if you could share your thoughts on whether these updates adequately address your concerns or if there are any additional points you would like us to clarify.
>
> Thank you once again for your thoughtful and constructive feedback.
>
> Best regards,
>
> The Authors

---

> > ### Comment · Reviewer_2kwB · 2024-12-03
> >
> > The reviewer thanks the authors for their efforts. I've read the original papers of the two additional baselines. Although the agent networks inside are still very different compared to MA$^2$E, the two baselines are more representative than TransMix. I found that the results of the two baselines in Figure 24 are severely inferior to those in their original papers. Is this due to the framework changing from *pymarl* to *pymarl2*?

---

> > > ### Author Response · Authors · 2024-12-03
> > >
> > > Dear Reviewer,
> > >
> > > We sincerely appreciate your time and prompt response.
> > >
> > > To ensure fairness and consistency, we utilized the pymarl2 framework and aligned all hyperparameters with the values outlined in the table above (e.g., batch size = 128). We believe the observed differences may stem from these changes.
> > >
> > > If there are any aspects requiring further clarification or elaboration, we would be happy to address them.
> > >
> > > Once again, we deeply appreciate your time and effort in reviewing our work.
> > >
> > > Best Regards,
> > >
> > > The Authors

---

### Official Review · Reviewer_bnEg · 2024-10-28

**Soundness:** 2
**Presentation:** 3
**Contribution:** 3
**Rating:** 6
**Confidence:** 3

**Summary:**

This paper presents Multi-Agent Masked Auto-Encoder (MA2E), which utilizes the masked auto-encoder architecture to infer the global state from partial observations for MARL. MA2E is trained by masking the agent-level information to recover the entire trajectories. Experiments on SMAC, SMACv2, and Google Research Football environments show that MA2E achieves superior performance when compared with representative MARL methods.

**Strengths:**

1.	The motivation for using MAE to infer the global state from agent observations is straightforward and well-motivated.
2.	The performance of the proposed MA2E is superior across different environments.
3.	The authors compare MA2E with other works using masked modeling such as MA2CL.

**Weaknesses:**

1.	Although it is claimed that MA2E can be easily integrated into existing MARL algorithms. Only QMIX-style algorithms are tested.
2.	MA2E is difficult to scale to a varying number of agents.

**Questions:**

1.	Why the results of COLA and MA2CL are missing in 10gen_terran and 10gen_protoss in Figure 4?
2.	Could the authors also compare with MaskMA? Could MA2E be extended into MAPPO and MADDPG?
3.	How does the sight range of each agent affect the performance of MA2E?

---

> ### Author Response · Authors · 2024-11-21
>
> Dear reviewer, we sincerely appreciate your careful review and valuable comments. We are also glad to have the opportunity to respond to your feedback. Our responses to the weaknesses and questions are presented below.
>
> ---
>
> **Weakness 1: Although it is claimed that MA²E can be easily integrated into existing MARL algorithms. Only QMIX-style algorithms are tested.**
>
> Please refer to Figure 7 and Figure 12, where we demonstrate the application of MA²E to RIIT [1] and IPPO [2], which are actor-critic style (policy-based) algorithms. These results show that integrating MA²E also improves performance in these methods.
> In general, CTDE MARL algorithms can be categorized into two main approaches: value-based methods with value decomposition (e.g., VDN, QMIX) and policy-based methods with actor-critic. As illustrated in Figure 3 and line 209-233, MA²E can be seamlessly integrated into each agent for policy-based algorithms without altering the backbone algorithm itself. The outputs of the backbone network and MA²E are generated independently and then aggregated, ensuring compatibility with a wide range of algorithms.
>
> ---
>
> **Weakness 2: MA²E is difficult to scale to a varying number of agents.**
>
> In our experiments, MA²E scaled from scenarios with two agents (e.g., 2s_vs_1sc in Figure 11) to ten agents (e.g., MMM2 in Figure 4). We want to note that the observation size also increase with the total entities in the environment (including enemies). Even with the large number of entities, we observed performance improvements. (30 entities in corridor and 66 entities in 2c_vs_64zg).
> Since input size is linear to the number of agents $n$ and we adopt the transformer architecture for MA²E, the complexity is $O(n^2)$. While this may limit scalability to a very large number of agents, it is comparable to other MARL models adopting transformer architectures [3-6], which exhibit a time complexity of at least $O(n^2)$. This demonstrates that MA²E does not have significant disadvantages in scalability compared to similar approaches in MARL.
> Scaling to a large number of agents remains a fundamental challenge for MARL algorithms, including our baselines. One potential solution would be leveraging permutation-invariance and equivariance [7, 8] to reduce the size of state space. Our future work will investigate scaling ability of MA²E and a way to manage its input size through such strategies.
>
> ---
>
> **Question 1: Why the results of COLA and MA2CL are missing in 10gen_terran and 10gen_protoss in Figure 4?**
>
> As SMACv2 is a recently released benchmark, most baselines have not been tested on it. We made efforts to modify the code to run on SMACv2 and included the available results. To avoid confusion, we separated the results for SMAC and SMACv2 into Figure 4 and Figure 5.
> Specifically, MA2CL is implemented using authors’ custom code rather than based on widely adopted frameworks like pymarl, and SMAC environment file was also modified. These factors made it challenging to conduct experiment in SMACv2.
> We are currently preparing additional experiments for COLA in SMACv2. If time permits, we will update the results as soon as possible.
>
> ---
>
> **Question 2: Could the authors also compare with MaskMA? Could MA²E be extended into MAPPO and MADDPG?**
>
> Regarding MaskMA, the experiments were conducted using offline expert datasets, which unfortunately are not publicly available. Since our study employs an online learning approach, the experimental settings are inconsistent, and the available results are not directly comparable. We carefully reviewed the [authors’ implementation](https://openreview.net/forum?id=Susy8EAff9), but it only includes the code for the MaskMA module itself. Details on how it interacts with the environment and how information is processed are not explicitly provided. Additionally, while the paper mentions that ‘For decentralized execution, we modify the attention matrix to ensure that each agent focuses solely on the surrounding agents during the self-attention process,’ the exact mask used during execution cannot be clearly identified. Despite our efforts, we were unfortunately unable to perform experiments on MaskMA due to these challenges.
> Since MAPPO and MADDPG are actor-critic style algorithms, MA²E can be integrated into into them, as explained in our respones to **Weakness 1**.

---

> ### Author Response · Authors · 2024-11-21
>
> **Question 3: How does the sight range of each agent affect the performance of MA²E?**
>
> The sight range of each agent affect its observation range, which in turn impacts the observation overlap between agents. As the sight range increases, observation overlap between agents becomes more significant.
> Intuitively, if the observation overlap is substantial, the information gathered by the agents is likely to be similar, reducing the impact of MA²E's inference capabilities. Conversely, when there is little or no overlap, it would become challenging for MA²E to reconstruct and infer effectively. Therefore, an appropriate level of overlap would be beneficial for leveraging MA²E effectively.
>
> We investigated this factor in Appendix D and E. Please refer to Appendix D and Figure 13. In the surround scenario, where there is significant observation overlap among agents, the benefit of MA²E is limited. In contrast, in the reflect scenario, agents are reasonably spaced apart, resulting in an appropriate level of overlap. Hence, the improvement over the backbone algorithms is larger in *reflect* than *surround*.
>
> As noted in Appendix E, there is very little observation overlap in 1o_2r_vs_4r scenario, which explains why the performance improvement of MA²E is not significant in Figure 14 and 15. We believe this is a fundamental limitation of the cases where only the local information is available. While communication-based methods would achieve better results, direct information exchange requires additional communication costs and constraints. Furthermore, as additional information beyond local observations are used, it does not strictly follow the CTDE assumption. Hence, comparison with non-communication methods would be unfair.
>
> ---
>
> We hope our responses have addressed your concerns. If there are any remaining issues, we would be grateful and pleased to discuss them during the discussion period. Thank you once again for your time and invaluable feedback.
>
> [1] Hu, Jian, et al. "Rethinking the Implementation Tricks and Monotonicity Constraint in Cooperative Multi-agent Reinforcement Learning." The Second Blogpost Track at ICLR 2023.
>
> [2] De Witt, Christian Schroeder, et al. "Is independent learning all you need in the starcraft multi-agent challenge?." arXiv preprint arXiv:2011.09533 (2020).
>
> [3] Yang, Yaodong, et al. "Qatten: A general framework for cooperative multiagent reinforcement learning." arXiv preprint arXiv:2002.03939 (2020).
>
> [4] Iqbal, Shariq, and Fei Sha. "Actor-attention-critic for multi-agent reinforcement learning." International conference on machine learning. PMLR, 2019.
>
> [5] Wen, Muning, et al. "Multi-agent reinforcement learning is a sequence modeling problem." Advances in Neural Information Processing Systems 35 (2022): 16509-16521.
>
> [6] Hu, Zican, et al. "Attention-Guided Contrastive Role Representations for Multi-Agent Reinforcement Learning." arXiv preprint arXiv:2312.04819 (2023).
>
> [7] Jianye, H. A. O., et al. "Boosting multiagent reinforcement learning via permutation invariant and permutation equivariant networks." *The Eleventh International Conference on Learning Representations*. 2022.
>
> [8] Lin, Bor-Jiun, and Chun-Yi Lee. "HGAP: Boosting Permutation Invariant and Permutation Equivariant in Multi-Agent Reinforcement Learning via Graph Attention Network." *Forty-first International Conference on Machine Learning*.

---

> > ### Comment · Reviewer_bnEg · 2024-11-26
> >
> > The reviewer appreciates the authors' response. Some of my concerns are addressed. Additionally, I think the experimental results need further improvement such as running RIIT on more maps (now just 2 maps) as well as comparing other MARL baselines that utilize the MAE technique.

---

> > > ### Author Response · Authors · 2024-11-28
> > >
> > > Dear reviewer, we appreciate your valuable feedback. We present our responses to your remaining concerns below.
> > >
> > > ---
> > >
> > > **Additional Experiments with RIIT**
> > >
> > > Thank you for your thoughtful suggestion. We kindly request the reviewer to refer to Figure 12, which presents further results employing MA²E to diverse backbone algorithms, including RIIT. Moreover, we presented the results for RIIT, along with additional experiments, separately in Figure 26. Combined with the results in Figure 7, MA²E consistently demonstrates improved performance across diverse scenarios, further highlighting its effectiveness. We hope this additional analysis addresses your concerns.
> > >
> > > ---
> > >
> > > **Comparison with other MARL baselines that utilizing the MAE technique**
> > >
> > > In response to the reviewer’s suggestion, we revisited the literature to identify other methods in the MARL domain that employ the MAE technique. Unfortunately, we were unable to identify additional methods. Moreover, as noted in our response to Question 2, the official code for MaskMA is not feasible for conducting experiments. To address your concern, we conducted additional experiments with MA2CL, and the results are presented in Figure 18. Combined with the results in Figure 4, these results further demonstrate MA²E’s superior performance compared to MA2CL, which utilizes the MAE technique.
> > >
> > > ---
> > >
> > > **Evaluation of COLA in SMACv2**
> > >
> > > To further address Question 1, we conducted additional experiments evaluating COLA in the SMACv2 environment. Additionally, we newly included the `10gen_zerg` scenario and conducted further experiments for all baselines to provide further empirical evaluation. The results are presented in Figure 19, and Figure 5 has been updated accordingly. MA²E consistently outperforms COLA and other baselines in SMACv2, demonstrating its superior performance across diverse domains.
> > >
> > > ---
> > >
> > > Once again, we sincerely appreciate your insightful comments and the time you have dedicated to reviewing our work.

---

> ### Author Response · Authors · 2024-12-03
> **Kind Reminder: Discussion Deadline Approaching**
>
> Dear Reviewer bnEg,
>
> We hope this message finds you well. Thank you for your valuable feedback and insightful suggestions. In response, we have conducted additional experiments on RIIT and COLA, as well as provided our thoughts on other MARL baselines.
>
> As the discussion period is coming to a close, we would greatly appreciate it if you could let us know whether these updates address your concerns or if there are any additional points you would like us to clarify.
>
> Once again, thank you for your time and valuable input throughout this process.
>
> Best regards,
>
> The Authors

---

### Official Review · Reviewer_GwqF · 2024-11-04

**Soundness:** 2
**Presentation:** 3
**Contribution:** 3
**Rating:** 6
**Confidence:** 5

**Summary:**

This paper introduces the Multi-Agent Masked Auto-Encoder (MA$^2$E), which leverages the masked auto-encoder architecture to infer information about other agents from partial observations. Specifically, MA$^2$E masks out the trajectories of other agents at random, enabling the agent to non-centrally reconstruct information about other agents based solely on its own local observations. This approach enhances sample efficiency and final performance during training in Dec-POMDP settings. The authors conducted extensive experiments in the SMAC, SMACv2, and GRF environments, demonstrating the outstanding performance of MA$^2$E.

**Strengths:**

1. The authors provide the necessary code and hyperparameter details. A preliminary review did not reveal any issues. Sharing the original code significantly enhances the credibility of the experimental results presented in the paper.

2. The motivation behind the paper is clearly articulated: the goal is to non-centrally reconstruct information about other agents using only the local observations of the agent itself, thereby improving sample efficiency in training. Although the method is straightforward, the experimental results reveal significant performance differences between MA$^2$E and other algorithms, particularly in super hard scenarios.

3. The selection of baselines in the experimental section is representative, incorporating many classic and relevant algorithms. Additionally, the experimental environments are diverse, including SMAC, SMACv2, and GRF.

4. The ablation studies are comprehensive, covering nearly all relevant variations. Furthermore, the visualization of observation inference accuracy is effectively executed.

**Weaknesses:**

1. There are several typos throughout the paper. For instance, a sentence is repeated in line 255, and there is an unnumbered figure in line 907.

2. MA$^2$E employs a two-stage rather than an end-to-end training approach, which contrasts with many existing works. While the authors demonstrate through ablation experiments that end-to-end training could lead to decreased performance, the pre-training aspect of the masked auto-encoder should still be factored into the algorithm's sample efficiency.

3. Based on the experiments, MA$^2$E still shows a significant correlation with observation overlap, indicating that communication-based methods cannot yet be fully replaced by MA$^2$E.

**Questions:**

1. Why did you choose to use trajectories from multiple time steps as input for the masked auto-encoder instead of the hidden states outputted by the RNN in the agent network?

2. SMAC has many representative scenarios, such as 8m_vs_9m, 10m_vs_11m, 27m_vs_30m, and 3s5z_vs_3s6z. Why were these scenarios not included or showcased in the experimental section?

3. Table 2 only presents the final convergence results of communication-based methods. Could you also provide their learning curves?

4. Similarly, regarding SMACv2, the paper only displays results for 10gen_terran (Terran_5_vs_5) and Protoss_5_vs_5. Can you include additional results from other scenarios?

---

> ### Author Response · Authors · 2024-11-21
>
> Dear reviewer, we deeply appreciate your thorough review and constructive comments. We are also grateful for the opportunity to respond to your feedback. Our responses to the weaknesses and answers to the questions are presented below.
>
> ---
>
> **Weakness 1: There are several typos throughout the paper. For instance, a sentence is repeated in line 255, and there is an unnumbered figure in line 907.**
>
> Thank you for pointing out the typos. We removed the repeated sentence (now at line 235 in the updated manuscript) and corrected the figure reference (now at line 919 in the updated manuscript). Additionally, we thoroughly proofread our paper to ensure there are no remaining typos or incorrect references. All updates have been highlighted in blue color.
>
> ---
>
> **Weakness 2:  MA²E employs a two-stage rather than an end-to-end training approach, which contrasts with many existing works. While the authors demonstrate through ablation experiments that end-to-end training could lead to decreased performance, the pre-training aspect of the masked auto-encoder should still be factored into the algorithm's sample efficiency.**
>
> While many existing studies adopt end-to-end learning methods, there are also advantages to independent learning. As shown in [1], independent learning is often employed when networks serve different roles. In the case of MA²E, its role is to effectively reconstruct the global state. Meanwhile, the role of the individual network (backbone) is generating Q-values or finding the optimal policy. Therefore, training them independently helps to avoid mutual interference, potentially leading to better performance. Moreover, independently trained MA²E demonstrates transferability by maintaining its performance (Figure 8 and lines 424-431). Nevertheless, as the reviewer suggested, there is a possibility that performance could improve with end-to-end training. Therefore, we are currently preparing additional experiments under an end-to-end setup (not completed due to limited time and resources). If these experiments are completed during the discussion period, we will provide an update accordingly.
>
> ---
>
> **Weakness 3: Based on the experiments, MA²E still shows a significant correlation with observation overlap, indicating that communication-based methods cannot yet be fully replaced by MA²E.**
>
> We would like to clarify that the primary focus of our paper is not to fully replace communication-based methods with MA²E. Instead, our objective is to address the challenge of partial observability while strictly following CTDE principle. Please refer to Table 2 and Figure 19. While the agents have the same observation overlaps in the same scenario, MA²E achieves better results than communication-based methods. Communication-based methods may perform better in cases with little or no observation overlap, however, agents use additional information beyond their local observations by directly exchanging information during execution. This deviates from the principle of decentralized execution (independent decision-making) and thus does not strictly align with CTDE [2]. Hence, comparison with non-communication methods should consider this difference. Furthermore, direct information exchange would be prohibited or not possible depending on target tasks, and communication requires additional cost and constraints. We believe that MA²E showed its effectiveness for the cases where only local information is available during execution.
>
> ---
>
> **Question 1: Why did you choose to use trajectories from multiple time steps as input for the masked auto-encoder instead of the hidden states outputted by the RNN in the agent network?**
>
> If we use RNN hidden states of the agent backbone network as input for MA²E, then MA²E would be dependent on the agent network. As the representation of hidden states changes as policy (backbone network) training progresses, the reconstruction target of MA²E varies. Since MA²E becomes dependent on agents’ internal hidden states, it would hinder independent training of MA²E, and the policy and MA²E training would be unstable. Furthermore, flexible integration of MA²E would become difficult. Using trajectories as input allows MA²E to reconstruct the information without aforementioned issues, as it can focus on environment dynamics. Nonetheless, we believe that utilizing RNN hidden states could be an interesting research direction and we will consider it for our future work.

---

> ### Author Response · Authors · 2024-11-21
>
> **Question 2: SMAC has many representative scenarios, such as 8m_vs_9m, 10m_vs_11m, 27m_vs_30m, and 3s5z_vs_3s6z. Why were these scenarios not included or showcased in the experimental section?**
>
> While SMAC environment offers a variety of scenarios, we adopted several Hard and Super-Hard scenarios from the original SMAC paper to demonstrate the effectiveness of MA²E, rather than easy scenarios (e.g., 10m_vs_11m). Nonetheless, we also believe that evaluation in additional scenarios could further strengthen our work.
> Please refer to Figure 18, where we present results from additional experiments. Due to the limited time and resources, we selected 8m_vs_9m and 3s5z_vs_3s6z scenarios, which have distinct characteristics (only marine entities and heterogeneous entities, respectively). While 3s5z_vs_3s6z is challenging for all algorithms, MA²E outperforms other baselines in 8m_vs_9m, demonstrating the effectiveness of our method.
>
> ---
>
> **Question 3: Table 2 only presents the final convergence results of communication-based methods. Could you also provide their learning curves?**
>
> Please refer to Figure 20 in the paper. We included the learning curves corresponding to the results in Table 2. While preparing this figure, we discovered that several log files were corrupted. We re-trained the affected cases and updated Table 2 accordingly (highlighted in blue). As shown in Figure 20, MA²E outperforms the baselines in most scenarios, further demonstrating its superiority of MA²E over communication-based methods.
>
> ---
>
> **Question 4: Similarly, regarding SMACv2, the paper only displays results for 10gen_terran (Terran_5_vs_5) and Protoss_5_vs_5. Can you include additional results from other scenarios?**
>
> Please refer to Figure 5 and 19 in the paper. We conducted additional experiments in 10gen_zerg scenario. Combined with the results in Figure 7 and 12, these experiments integrating MA²E consistently enhances performance within SMACv2 environment.
>
> ---
>
> We hope our responses have addressed your concerns. If there are any remaining issues, we would be thankful if you could discuss them with us during the discussion period. Thank you once again for your time and valuable input.
>
> [1] Venkatraman, Siddarth, et al. "Reasoning with latent diffusion in offline reinforcement learning." *arXiv preprint arXiv:2309.06599* (2023).
>
> [2] Liu, Bo, et al. "Coach-player multi-agent reinforcement learning for dynamic team composition." *International Conference on Machine Learning*. PMLR, 2021.

---

> ### Author Response · Authors · 2024-11-28
> **Kind Reminder: Feedback on Our Responses**
>
> Dear Reviewer GwqF,
>
> We hope this message finds you well. We sincerely thank you for your time and efforts in reviewing our paper.
>
> This is a friendly reminder to kindly provide your feedback on our responses. We would greatly appreciate it if you could share your opinions or let us know if there are any additional steps required from our side.
>
> Once again, thank you for your time and attention.
>
> Best regards,
>
> The Authors

---

> ### Author Response · Authors · 2024-12-03
> **Kind Reminder: Discussion Deadline Approaching**
>
> Dear Reviewer GwqF,
>
> We hope this message finds you well. Your feedback is invaluable to strengthening our research, and we greatly appreciate the time and effort you have already dedicated.
>
> As the discussion period approaches its conclusion, we would deeply appreciate it if you could share your thoughts.
>
> Once again, thank you for your time and consideration.
>
> Best regards,
>
> The Authors

---

### Official Review · Reviewer_dk6X · 2024-11-05

**Soundness:** 3
**Presentation:** 2
**Contribution:** 3
**Rating:** 8
**Confidence:** 4

**Summary:**

Despite the advancements achieved with CTDE, partial observability remains a barrier to effective cooperation among agents. This paper's approach is to enable each agent to infer global information solely from its local observations, allowing it to gain a comprehensive view without relying on communication. To achieve this, the authors introduce the Multi-Agent Masked Auto-Encoder, which leverages the masked auto-encoder architecture to infer the information of other agents from partial observations. By using masking to learn how to reconstruct global information, the method functions as an inference module for individual agents within the CTDE framework. The method can be integrated into existing multi-agent reinforcement learning (MARL) algorithms and has been experimentally shown to enhance performance across a variety of environments.

**Strengths:**

- The paper addresses an important problem in cooperative MARL, that is dealing with partial observability without communication.
- The paper proposes an interesting, novel method which achieves very good performance in challenging SMAC and SMACv2 tasks, outperforming well-known MARL baselines, such as QMIX, QPLEX, using a relatively small number of training timesteps.
- The proposed method outperforms methods based on similar approaches, such as MAC2L [1].
- Interestingly, the proposed method can achieve comparable performance, or even outperform state-of-the-art communication-based methods.

[1] Song, Haolin, et al. "MA2CL: masked attentive contrastive learning for multi-agent reinforcement learning." Proceedings of the Thirty-Second International Joint Conference on Artificial Intelligence. 2023.

**Weaknesses:**

- The framework requires a pre-training stage, which can be impractical for certain scenarios. How much pre-training is needed? How extra wall-clock time is needed for the pre-training?
- The paper conducts a limited number of experiments in GRF (that is, only for a comparison with VDN).
- The presentation in the incorporation of the method into the MARL backbone algorithm can be improved.

The authors have addressed my concerns.

**Questions:**

- CDS [1], a well-known MARL paper, has reported a better performance (than MA$^2$E), using the same number of training time steps, in super-hard corridor, MMM2 and 6h vs 8z tasks, in which this paper reports a "more pronounced improvement" over baselines. Therefore, it would be beneficial if the authors included in the comparison a few exploration-based state-of-the-art methods (such as CDS) which have shown remarkable performance in SMAC tasks.
- In execution time, for agent $i$, the paper says that MA$^2$E takes only the local information of agent $i$. However, in Fig. 2, it is illustrated that the model should take as input information of all agents. So, in execution time, does the model simply masks the positions (trajectories) of the other agents?
- Considering the strong performance of the proposed method, as compared with the communication-based methods, have the authors compared the method with the communication-based baselines in Traffic Junction benchmark (which provides very challenging tasks aiming to test the effectiveness of agent communication)?

[1] Li, Chenghao, et al. "Celebrating diversity in shared multi-agent reinforcement learning." Advances in Neural Information Processing Systems 34 (2021): 3991-4002.

---

> ### Author Response · Authors · 2024-11-21
>
> Dear reviewer, we are sincerely grateful for your comprehensive feedback and constructive comments. We also appreciate the opportunity to respond to your concerns. Our responses are presented below.
>
> ---
>
> **Weakness 1: The framework requires a pre-training stage, which can be impractical for certain scenarios. How much pre-training is needed? How extra wall-clock time is needed for the pre-training?**
>
> Please refer to Table 8 and Table 9 in the paper, where we analyze the training costs of MA²E+QMIX compared to the backbone QMIX method in two scenarios. As shown in the tables, the pre-training times required for MA²E are relatively small, accounting for only 4.5% and 10.5% of the total time to convergence, respectively.
> While applying MA²E to QMIX introduces a pre-training stage, MA²E+QMIX converges faster than QMIX. This demonstrates that MA²E improves sample efficiency and accelerates convergence speed in terms of both time and steps. As observed in our experiments, these enhanced sample efficiency lead to shorter overall training times and improved performance.
> We also report the measured pre-training times below:
> | Scenario | 5m_vs_6m | 2c_vs_64zg | 3s_vs_5z | corridor | 6h_vs_8z | MMM2 |
> | --- | --- | --- | --- | --- | --- | --- |
> | Pre-training time | 0.65h | 0.84h | 0.86h | 1.10h | 1.02h | 0.93h |
>
> ---
>
> **Weakness 2: The paper conducts a limited number of experiments in GRF (that is, only for a comparison with VDN).**
>
> While we explored the extensibility of MA²E in various ways throughout the paper, we agree that the additional evaluations in the GRF environment would further strengthen our work. We are currently preparing additional experiments but were unable to complete them due to limited time and resources. If time permits, we will update the results during discussion period. As noted in the [pymarl2 repository](https://github.com/hijkzzz/pymarl2/tree/master), the original GRF environment does not provide global state information and only offers observation representations. Hence, many backbone algorithms, such as QMIX, are not directly applicable in GRF. To address this issue, we are exploring two strategies: integrating MA²E with IPPO, which is not affected by the lack of global state information, or modifying the GRF environment to facilitate access to global state.
>
> ---
>
> **Weakness 3: The presentation in the incorporation of the method into the MARL backbone algorithm can be improved.**
>
> Thank you for your feedback on the presentation. We have revised the writing to improve clarity and better illustrate how MA²E is integrated into the MARL backbone algorithm. The updated content is highlighted in blue (line 209-215).
>
> ---
>
> **Question 1: CDS, a well-known MARL paper, has reported a better performance (than MA²E), using the same number of training time steps, in super-hard corridor, MMM2 and 6h vs 8z tasks, in which this paper reports a "more pronounced improvement" over baselines. Therefore, it would be beneficial if the authors included in the comparison a few exploration-based state-of-the-art methods (such as CDS) which have shown remarkable performance in SMAC tasks.**
>
> Thank you for suggesting a new baseline. We also believe that comparison with exploration-based methods would be beneficial. Please refer to Figure 17 and Table 6 in the paper. We conducted additional experiments with CDS. Furthermore, we included additional baselines, RODE and LDSA, as suggested by Reviewer 2kwB. While CDS and other baselines fails to train in certain scenarios, MA²E consistently learns and improves performance across all scenarios. Furthermore, as shown in Table 6, QMIX+MA²E demonstrates the best average performance. These results further validate the effectiveness and robustness of our method.
>
> ---
>
> **Question 2: In execution time, for agent, the paper says that MA²E takes only the local information of agent  However, in Fig. 2, it is illustrated that the model should take as input information of all agents. So, in execution time, does the model simply masks the positions (trajectories) of the other agents?**
>
> The reviewer’s interpretation is correct. Please refer to line 193-198 and 209-214 in the paper. During execution, the trajectories of all other agents are masked, and only the local information of the current agent remains. MA²E reconstructs trajectories of other agents based solely on local information. Figure 2 illustrates the architecture of MA²E and the masking process. To enhance clarity, we have updated the caption of Figure 2 to explain how MA²E operates during decentralized execution.

---

> ### Author Response · Authors · 2024-11-21
>
> **Question 3: Considering the strong performance of the proposed method, as compared with the communication-based methods, have the authors compared the method with the communication-based baselines in Traffic Junction benchmark (which provides very challenging tasks aiming to test the effectiveness of agent communication)?**
>
> Thank you for suggesting these insightful tasks. While a direct comparison with communication-based methods is not the primary focus of our paper, evaluating our method in a traffic junction domain would be an interesting research direction. Unfortunately, due to constraints in time and computational resources, as well as our limited familiarity with such environments, we are unable to conduct this experiment at present. As there are a few relevant traffic control benchmarks available, we would greatly appreciate it if you could recommend one. We will carefully consider it for inclusion in our paper or as part of future work.
>
> ---
>
> We hope our responses have addressed your concerns. If there are any remaining issues, we would be grateful if you could discuss them with us during the discussion period. Thank you once again for your time and effort.
>
> [1] Hu, Jian, et al. "Rethinking the Implementation Tricks and Monotonicity Constraint in Cooperative Multi-agent Reinforcement Learning." The Second Blogpost Track at ICLR 2023.

---

> > ### Comment · Reviewer_dk6X · 2024-11-21
> >
> > I would like to thank the authors for addressing my concerns. I propose the authors include the extra experiments in the main body of the paper. I am happy to **raise my score to 8**.
> >
> > Regarding the traffic junction benchmark, I was referring to the one used in [1]. From my personal experience, this is a very challenging benchmark requiring excessive agent communication. It would be very interesting to explore the proposed method on this benchmark as well.
> >
> > [1] Guan, Cong, et al. "Efficient multi-agent communication via self-supervised information aggregation." Advances in Neural Information Processing Systems 35 (2022): 1020-1033.

---

> > > ### Author Response · Authors · 2024-11-22
> > >
> > > Dear Reviewer dk6X,
> > >
> > > We sincerely appreciate your support and constructive feedback. Following your suggestion, we will include the additional experiments in the main body of the paper. As some additional experiments are in progress, we will ensure the updates are incorporated into a later version of the paper. Additionally, we will carefully explore the traffic junction benchmark you recommended.
> > >
> > > Thank you again for your time and invaluable comments.
> > >
> > > Best regards,
> > >
> > > The Authors

---

### Author Response · Authors · 2024-11-21
**Common Response**

Dear reviewers, we would like to express our gratitude for your invaluable feedback on our paper. In response to the reviewers’ comments, we conducted additional experiments and updated our paper. The updated sections are highlighted in blue for your convenience.

- **Additional Experiment and Results**
    - To enhance the clarity of comparison, we created **Appendix H** to include additional experiments, starting from **page 21**.
    - **[Baselines and Scenarios]** We added **CDS** [1], **LDSA** [2], **RODE** [3] as new baselines, as reviewer dk6X and 2kwB suggested. We also added **8m_vs_9m, 3s5z_vs_3s6z, 10gen_zerg** as reviewer GwqF suggested. Please refer to **Appendix H.1 and H.2** The results are illustrated in **Figure 17, 18, 19 and Table 6**.
    Below we provide a summary of the additional experiments (◎: new, O: already done in paper).


        |  | 3s_vs_5z | 2c_vs_64zg | 3s_vs_5z | corridor | 6h_vs_8z | MMM2 | 8m_vs_9m | 3s5z_vs_3s6z |
        | --- | --- | --- | --- | --- | --- | --- | --- | --- |
        | LDSA | ◎ | ◎ | ◎ | ◎ | ◎ | ◎ | ◎ | ◎ |
        | CDS | ◎ | ◎ | ◎ | ◎ | ◎ | ◎ | - | - |
        | RODE | ◎ | ◎ | ◎ | ◎ | ◎ | ◎ | - | - |
        | COLA | O | O | O | O | O | O | ◎ | ◎ |
        | MA2CL | O | O | O | O | O | O | ◎ | ◎ |

        |  | QMIX+MA²E(ours) | QMIX | VDN | OW-QMIX | CW-QMIX | QPLEX |
        | --- | --- | --- | --- | --- | --- | --- |
        | 10gen_zerg | ◎ | ◎ | ◎ | ◎ | ◎ | ◎ |
    - **[Learning curves of Table 2]** We added the learning curves of the results in Table 2, as reviewer GwqF suggested. Please refer to **Appendix H.3 and Figure 20**. While preparing this figure, we discovered that several log files were corrupted. We re-trained the affected cases and updated Table 2 accordingly.
    - **[Ablation studies]** We conducted additional ablation studies, as reviewer 2kwB suggested. Please refer to **Appendix H.4 and Figure 21, 22**.
    - **[Analysis on Training Costs]** We analyzed the training costs in **Appendix H.5**. Please refer to **Table 8 and 9.**

- **Writing**
    - We corrected the typos, including mentioned by reviewer GwqF, and updated our paper following comprehensive proofreading.
    - We improved ‘Incorporating MA²E into Individual Agent’ part in Section 3 (line 209-215) as suggested by reviewer dk6X.

If there are any questions or concerns, we would be grateful if you could share them during the discussion period.

---

[1] Li, Chenghao, et al. "Celebrating diversity in shared multi-agent reinforcement learning." Advances in Neural Information Processing Systems 34 (2021): 3991-4002.

[2] Yang, Mingyu, et al. "Ldsa: Learning dynamic subtask assignment in cooperative multi-agent reinforcement learning." *Advances in Neural Information Processing Systems* 35 (2022): 1698-1710.

[3] Wang, Tonghan, et al. "Rode: Learning roles to decompose multi-agent tasks." *arXiv preprint arXiv:2010.01523* (2020).

---

### Meta-Review · Area_Chair_h1pe · 2024-12-19

**Metareview:**

This paper proposes the Multi-Agent Masked Auto-Encoder (MA2E), which utilizes the masked auto-encoder architecture to infer the information of other agents from partial observations. All reviewers believe that the motivation of this paper is clear, the method is novel, and the experiments are thorough. The main point of contention regarding this paper is the comparison between the ablation study setup and the baseline experiments. AC reviewed all of the authors' rebuttals and, considering that the authors provided all reproducible code, the AC believes they are sufficient and therefore proposes to accept this paper.

**Additional Comments On Reviewer Discussion:**

The focus of the discussion between the authors and the reviewers during the rebuttal period was on the supplementation and analysis of the experiments. The authors made significant efforts to address all the reviewers' concerns. Their efforts led one reviewer to raise their score to 8, and they continuously responded to the reviewers' questions over multiple rounds. AC believes that the authors have effectively addressed the reviewers' concerns.

---

### Decision · Program_Chairs · 2025-01-22

Accept (Poster)